# Effect of Aquatic Vegetation Restoration after Removal of Culture Purse Seine on Phytoplankton Community Structure in Caizi Lakes

Wenqian Zhao, Zhenzhong Liu, Wenli Guo and Zhongze Zhou *

School of Resources and Environmental Engineering, Anhui University, Hefei 230601, China;
x19201007@stu.ahu.edu.cn (W.Z.); x19301076@stu.ahu.edu.cn (Z.L.); x19301100@stu.ahu.edu.cn (W.G.)
* Correspondence: zhzz@ahu.edu.cn

**Abstract:** Many reports have demonstrated that the removal of aquaculture purse seine is conducive to the restoration of aquatic vegetation and the improvement in water quality, but less attention has been paid to the effects on phytoplankton. This paper addressed the response of phytoplankton community structure to aquatic vegetation restoration after purse seine removal in Caizi Lakes. The results showed that the average dissolved oxygen (from $7.43 \pm 0.25$ mg/L to $9.12 \pm 0.49$ mg/L) and Secchi depth (from $28.40 \pm 6.20$ cm to $47.61 \pm 14.62$ cm) in the water column of the Caizi Lakes increased after the restoration of aquatic vegetation, while the average concentrations of total nitrogen (from $2.00 \pm 0.16$ mg/L to $1.34 \pm 0.18$ mg/L) and total phosphorus (from $0.15 \pm 0.02$ mg/L to $0.06 \pm 0.01$ mg/L) decreased. After the restoration of the aquatic vegetation, the cell density of phytoplankton declined from $21.04 \pm 4.57 \times 10^6$ cells/L to $12.74 \pm 3.63 \times 10^6$ cells/L, and the biomass fell from $18.13 \pm 3.57$ mg/L to $9.72 \pm 2.55$ mg/L. We also observed that Shannon–Wiener diversity, Margalef and Pielou indices of phytoplankton surged by 66.50%, 46.20% and 84.68%, respectively. Because this study demonstrated that aquatic vegetation could alleviate the eutrophication, it can provide guidance for the restoration and protection of the aquatic ecosystem.

**Keywords:** purse seine removal; restoration of the aquatic vegetation; phytoplankton community structure; phytoplankton diversity





## 1. Introduction

The middle and lower reaches of the Yangtze River is the most concentrated area of freshwater resources in China, and a large area of shallow lakes are distributed on both sides of the Yangtze River. Together with the Yangtze River, these lakes constitute a unique river–lake complex ecosystem and provide good habitat conditions for maintaining biodiversity [1]. Since 2010, purse seine culture in lakes connected to the Yangtze River has increased dramatically; 80% of the water surface has been covered by purse seine. The increased density of fish, such as herbivorous fish, has threatened the survival of aquatic vegetation, especially the severe degradation of submerged vegetation [2]. The unused bait from the farming process decomposes into the water column or settles on the lake bottom, causing persistent pollution to the environment [3]. In addition, farming pollution has been superimposed within the seine area due to the barrier effect of the seine. Therefore, prolonged high-density seine farming will cause eutrophication [4]. In order to restore the structure and function of the aquatic ecosystem of the Yangtze River-connected Lake, relevant governments have dismantled the aquaculture purse seine. For example, the Anqing municipal government completely dismantled the water surface purse seine of the Caizi Lakes in October 2017 and cleared the water surface purse seine of 58 km² and about 270 km² in April 2018, thus achieving the "three noes" goal of no purse seine, no fishing boats and no fishermen in the lake.

Numerous studies have shown that the removal of aquaculture seines is beneficial to the restoration of aquatic vegetation diversity. Some scholars found that the removal of purse seine led to a shift in the dominant population of aquatic vegetation in East Lake Taihu, from submerged plants to floating-leaved plants, based on remote sensing monitoring [5]. Others observed that aquatic vegetation such as *Trapa bispinosa*, *Nymphoides peltatum* and *Zizania latifolia* recovered rapidly, and the number and cover of aquatic vegetation species increased significantly after the removal of aquaculture seine in Huayang Lakes [6]. Aquatic vegetation is of great significance to the stability of the aquatic ecosystem. It is not only a key link in the food chain of the aquatic ecosystem, but also increases the water transparency, stabilizes sediment and reduces water velocity [7].

Like large aquatic vegetation, as an important part of aquatic ecosystem, phytoplankton also plays a vital role in biogeochemical cycles [8]. There are complex interrelationships between aquatic vegetation and phytoplankton, such as competition [9] and allelopathy [10]. It was found that the restoration of the aquatic vegetation caused the change in phytoplankton community structure. Aquatic vegetation can inhibit the growth of phytoplankton by competing for light, nutrients and living space [11,12]. In addition, it can also inhibit phytoplankton growth by secreting certain specific chemicals through allelopathy [13]. Some scholars pointed out that allelochemicals produced by *Potamogeton crispus* significantly inhibited the growth of *Scenedesmus* [14]. Others uncovered that *Nymphaea lotus* and *Polygonum limbatum* contain phenolic and carboxyl compounds which can inhibit phytoplankton reproduction [15]. Our investigation revealed that floating plants, mainly *T. bispinosa* and *Eichhornia crassipes*, recovered rapidly after the removal of the aquaculture seine in Caizi Lakes, especially in Lake Baitu, where 40% of the water surface was covered by aquatic vegetation. In view of the key role of aquatic vegetation in aquatic systems and its relationship with phytoplankton, we hypothesized that the change in phytoplankton community structure after the removal of the seine was caused by the restoration of aquatic vegetation.

In order to verify this hypothesis, we analyzed the phytoplankton community structure, the aquatic vegetation and the water quality of the Caizi Lakes before and after the aquatic vegetation restoration. The main issues addressed are as follows: (1) determine the restoration status of aquatic vegetation after purse seine removal; (2) analyze the influence of aquatic vegetation restoration on lake water quality; (3) explore the influence of water quality and aquatic vegetation changes on the phytoplankton community structure before and after the aquatic vegetation restoration. This study evaluates the significance of aquatic vegetation restoration after the purse seine removal to the lake ecosystem from the perspective of phytoplankton, which can indicate the protection and ecological restoration of the lake.

## 2. Materials and Methods

### 2.1. Description of Study Area

The Caizi Lakes (30°42′–30°58′ N, 117°01′–117°09′ E) are located on the north bank of the middle and lower reaches of the Yangtze River, composed of Lake Baitu, Lake Caizi and Lake Xizi, and are typical shallow river-connecting lakes [16]. The lakes area is 242.3 km$^2$ in the summer and shrinks to 145.2 km$^2$ in the winter due to a decrease in water level, with an average water depth of 1.7 m [17]. The shallow lakes are located in the subtropical monsoon climate zone, with an annual average precipitation of 1200–1389 mm, annual average temperature of 16.5 °C, summer average temperature of 29.0 °C and winter average temperature of 3.5 °C. Controlled by Zongyang Gate, the water level of the lakes is the highest in summer, and this decreases in the autumn and winter, with obvious dynamic changes in water level [18].

### 2.2. Sampling Point Setting and Sampling Time

We carried out 12 investigations and samples before and after the removal of the breeding purse seine (October 2016, January 2017, April 2017 and July 2017) and after



the removal of the purse seine (August 2019, October 2019, January 2020, April 2020 and July 2020). We set up 20 sample points in the whole lake, including 7 in Lake Baitu, 10 in Lake Caizi and 3 in Lake Xizi (Figure 1).

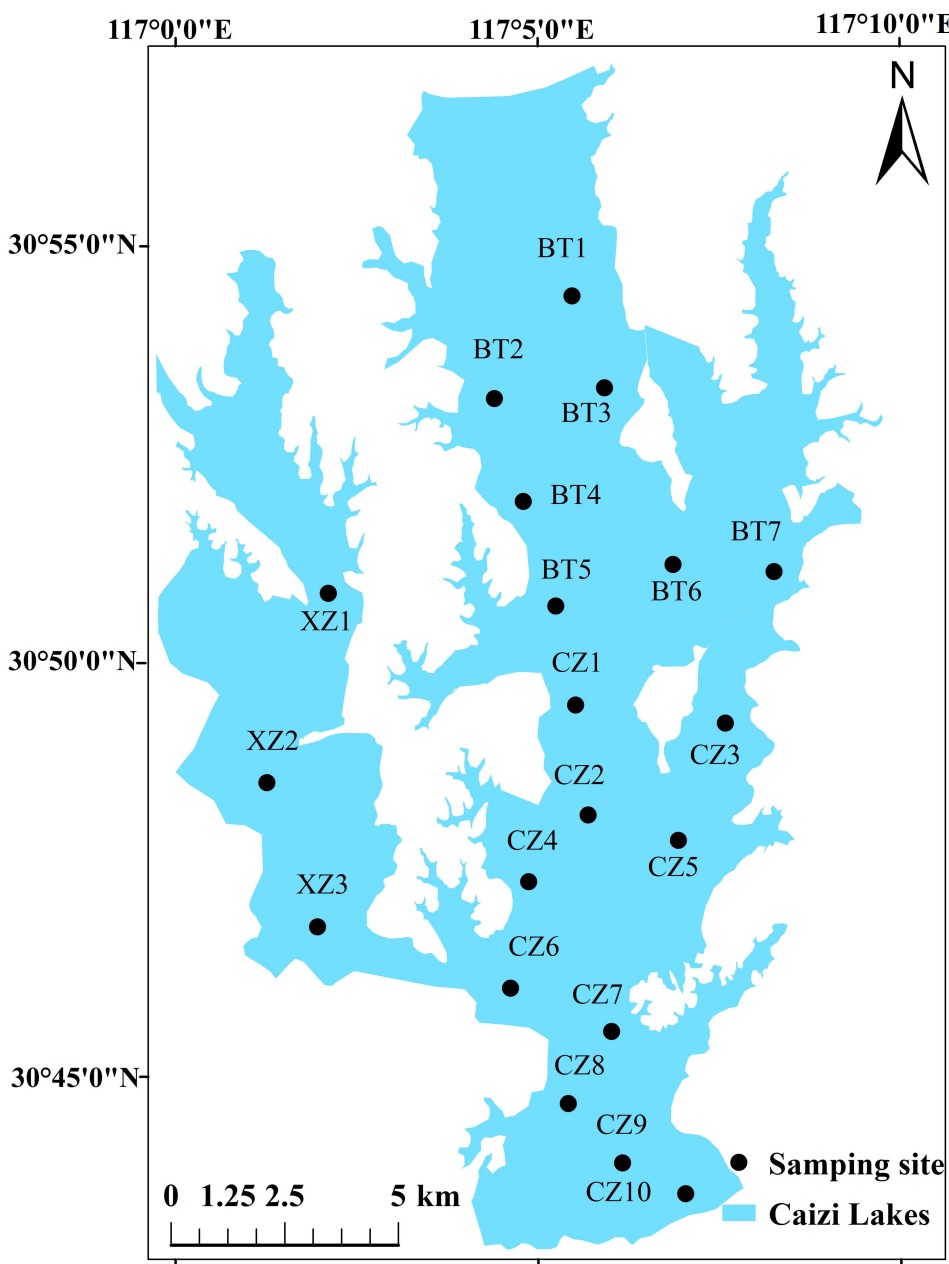

**Figure 1.** Distribution of sampling points in Caizi Lakes (BT to Baitu, CZ to Caizi, and XZ to Xizi).

The government carried out a complete removal of the farming seine in the Caizi Lakes from October 2017 to April 2018. A total of 12 surveys were conducted on the Caizi Lakes before (October 2016, January 2017, April 2017, and July 2017) and after (August 2019, October 2019, January 2020, April 2020, July 2020, October 2020, April 2021, and July 2021) the removal of the farming seine in this study sampling. A total of 20 sampling points were set up in the whole lake, including 7 in Lake Baitu, 10 in Lake Caizi and 3 in Lake Xizi (Figure 1).

*2.3. Data Collection*

We used the Hach HQ40d portable multimeter to measure water temperature (WT), dissolved oxygen (DO) and pH, the Hach 2100Q portable turbidimeter to measure tur-

bidity (Turb), and the Secchi disk to measure water transparency (SD) and water depth (WD). We collected 5 L mixed water samples at 0.5 m below the water surface with a cylindrical sampler. For nutrient concentration analysis, we collected 1 L water samples. To determine the chlorophyll *a* concentration, we used 1 L water samples. Magnesium carbonate solution was added to prevent pigment dissolution, and 1 L water samples were collected for phytoplankton identification and counting, and all of them were stored at 4 °C. We surveyed the species and distribution of aquatic plants in the Caizi Lakes by visual boating. In addition, we also collected remote sensing images from Sentinel-2 MSI of the Caizi Lakes in July 2017, August 2019 and July 2021 (http://slj.anqing.gov.cn, accessed on 27 July 2017, 21 August 2019, 30 July 2021). The image has less cloud cover, high quality and 10 m ground resolution. We processed remote sensing image data by The Environment for Visualizing Images (version 5.5), used the decision tree classification to distinguish vegetated areas from non-vegetated areas, and calculated the coverage of vegetated areas (Cov) [19]. The nutrients and chlorophyll *a* concentration (Chl. *a*) were analyzed in a laboratory in accordance with the national standard method SEPA [20]. Total nitrogen (TN) was determined by the alkaline potassium persulfate digestion UV spectrophotometric method, total phosphorus (TP) by the ammonium molybdate spectrophotometric method, ammonia nitrogen ($NH_4^+$-N) by Nessler's reagent spectrophotometry and nitrate nitrogen ($NO_3^-$-N) by UV spectrometry, and Chl. *a* by acetone extraction spectrophotometry under an ultraviolet spectrophotometer (UV 2450). After three replicates, we calculated the average values of the above indices. Quantitative phytoplankton samples were fixed with Lugol iodine solution at a fixation dose of 1% of the sample volume. We removed the supernatant slowly by siphoning after they had precipitated for at least 48 h until concentrated to 30 mL. After mixing, we took 0.1 mL samples and placed them in a plankton counting chamber [6], and randomly selected 100 visual fields under ×400 times optical microscope (BX53, Olympus) to classify and count the number of cells in phytoplankton [21].

*2.4. Data Processing*

We counted the number of phytoplankton cells and calculated their cell density and biomass. Biomass was obtained by multiplying the number of cells and the weight of cells. Since the specific gravity of phytoplankton is close to 1, we directly converted the volume to weight, and the cell volume was determined by the average cell dimensions of each species. The phytoplankton community structure was characterized by species dominance ($Y$) and biodiversity including the Shannon–Wiener index ($H'$) [22], Margalef index ($D$) [23], and Pielou index ($J$) [24]. The calculation formula for this is:

$$Y = n_i/N \times f_i \tag{1}$$

$$H' = -\sum_{i=1}^{s} (n_i/N) \times \log_2(n_i/N) \tag{2}$$

$$D = (S-1)/\log_2 N \tag{3}$$

$$J = H'/\log_2 S \tag{4}$$

where $n_i$ is the number of individuals of species $i$, $N$ is the total number of individuals of all species, $f_i$ is the frequency of individuals in the $i$ species, $S$ is the total number of phytoplankton species, and $n_i/N$ represents the relative proportion of species $i$. $Y \geq 0.02$ is the dominant species. $H'$ = 0~1 (heavy pollution type), 1~3 (moderate pollution type), >3 (light pollution or no pollution type); $D$ = 0~2 (heavy pollution type), 2~4 (medium pollution type), 4~6 (light pollution type), >6 (clean); $J$ = 0~0.3 (heavy pollution type), 0.3~0.5 (medium pollution type) and >0.5 (light pollution or no pollution type).

We used the comprehensive nutrient status index (*TLI*) to evaluate the nutrient status of the lakes, and take *Chl. a*, *TN*, *TP* and *SD* as the evaluation parameters to calculate the weights. The formula to calculate this is [25]:

$$TLI(\textstyle\sum) = \sum_{j=1}^{m} w_j \times TLI(j) \tag{5}$$

$$w_j = r^2{}_{ij} / \sum_{j=1}^{m} r^2{}_{ij} \tag{6}$$

In the formula, *TLI* (Σ) is the comprehensive nutritional status index; $w_j$ is the correlation weight of nutritional status parameter of *j*-th parameter; $r_{ij}$ is the *j*-th parameter and reference parameter; m is the number of evaluation parameters; *TLI* (*j*) is the nutritional state parameter representing the *j*-th parameter, which is calculated according to the following formula:

$$TLI(Chl.a) = 10(2.5 + 1.086 \ln Chl.a) \tag{7}$$

$$TLI(TP) = 10(9.436 + 1.624 \ln TP) \tag{8}$$

$$TLI(TN) = 10(5.453 + 1.694 \ln TN) \tag{9}$$

$$TLI(SD) = 10(5.118 - 1.94 \ln SD) \tag{10}$$

$$TLI(COD_{Mn}) = 10(0.109 + 2.661 \ln COD_{Mn}) \tag{11}$$

where *Chl. a* is chlorophyll *a* concentration (ug/L), *TP* is total phosphorus concentration (mg/L), *TN* is total nitrogen concentration (mg/L), *SD* is transparency (m), and $COD_{Mn}$ is permanganate index (mg/L).

We utilized SPSS software (26.0 version) to test the difference in various parameters of nonidentical samples by one-way ANOVA, and the Pearson correlation analysis was employed to evaluate the influence of environmental factors on cell density and biomass of phytoplankton. We used CANOCO (version 5.0) for redundancy analysis to evaluate the relationship between the biomass of dominant phytoplankton species and environmental factors. We used detrended correspondence analysis (DCA) to determine that the first axis gradient length was all less than 3, so we chose linear model-based RDA analysis. The phytoplankton community and the environmental parameters were tested by the Monte Carlo method. In order to eliminate the influence of extreme value, we transformed the data into log (x + 1) before statistical analysis to satisfy the normal distribution and tested the normal distribution by Shapiro–Wilk. In addition, we used SPSS to test the variance homogeneity of data. We utilized ArcGIS (version 10.6) to map the location of sampling sites and the distribution of aquatic vegetation, and Origin (version 2021) to draw histograms.

## 3. Results

### 3.1. Physical and Chemical Indicators of Water Body

In time, before and after the aquatic vegetation restoration, most of the physical and chemical factors of the Caizi Lakes have similar seasonal variation rules, but there are obvious differences in annual variation (Table 1). After the restoration of aquatic vegetation, the mean values of pH, DO and SD in the Caizi Lakes climbed significantly ($p < 0.05$), with a pH from 7.17 ± 0.14 to 8.30 ± 0.15 and DO from 7.43 ± 0.25 mg/L to 9.12 ± 0.49 mg/L. SD ranged from 28.40 ± 6.20 cm to 47.61 ± 14.62 cm. SD demonstrated an obvious seasonal variation after the restoration of the aquatic vegetation, and SD in the summer was significantly higher than that in other seasons ($p < 0.01$). After the restoration of aquatic vegetation, the average concentrations of TP, TN and AN declined significantly ($p < 0.01$), among which TP from 0.15 ± 0.02 mg/L to 0.06 ± 0.01 mg/L, TN from 2.00 ± 0.16 mg/L to 1.34 ± 0.18 mg/L, and AN from 0.61 mg/L ± 0.05 mg/L to 0.47 mg/L ± 0.02 mg/L. In terms of seasonal changes, TP concentration was highest in summer and lowest in

winter before and after aquatic vegetation restoration. However, TN concentration was higher in summer and autumn than in spring and winter before purse seine removal, and higher in spring and winter than in summer and autumn after recovery. After the restoration of aquatic vegetation, the concentration of Chl. *a* and Turb dropped significantly in the Caizi Lakes ($p < 0.05$). Before and after the fence was removed, the concentration of Chl. *a* was $6.10 \pm 2.36$ mg/L and $3.96 \pm 1.78$ mg/L, and Turb was $16.70 \pm 2.13$ NTU and $15.42 \pm 1.76$ NTU, respectively. In addition, Chl. *a* was higher in summer and autumn than in spring and winter (Figure 2k). The *TLI* of aquatic vegetation before and after restoration was $61.59 \pm 0.67$ and $53.58 \pm 0.71$, respectively, which indicated that the Caizi Lakes changed from moderate eutrophication to mild eutrophication (Figure 2l).

**Table 1.** Physical and chemical factors of water before and after aquatic vegetation restoration (average value ± standard deviation). Before stands for before recovery and After stands for after recovery. January is winter, April is spring, July and August are summer, and October is autumn.

| | | WT (°C) | WD (m) | pH | SD (cm) | DO (mg/L) | TP (mg/L) | TN (mg/L) | NH4$^+$-N (mg/L) | NO3$^-$-N (mg/L) | Turb (NTU) | COD$_{Mn}$ (mg/L) | Chl. *a* (ug/L) |
|---|---|---|---|---|---|---|---|---|---|---|---|---|---|
| Before | October 2016 | 16.9 ± 1.09 | 3.4 ± 0.16 | 7.23 ± 0.1 | 24.36 ± 1.63 | 7.34 ± 0.18 | 0.15 ± 0.02 | 2.1 ± 0.06 | 0.64 ± 0.03 | 0.34 ± 0.06 | 15.05 ± 0.37 | 12.31 ± 0.74 | 5.13 ± 0.36 |
| | January 2017 | 6.89 ± 0.41 | 1.3 ± 0.08 | 7.12 ± 0.05 | 24.79 ± 0.47 | 7.93 ± 0.03 | 0.13 ± 0.01 | 1.82 ± 0.08 | 0.52 ± 0.02 | 0.21 ± 0.01 | 16.68 ± 0.13 | 11.45 ± 0.42 | 4.27 ± 0.34 |
| | April 2017 | 17.52 ± 0.13 | 1.65 ± 0.04 | 6.98 ± 0.1 | 26.36 ± 0.02 | 7.29 ± 0.02 | 0.15 ± 0.01 | 1.87 ± 0.05 | 0.62 ± 0.02 | 0.33 ± 0.12 | 14.87 ± 0.42 | 12.02 ± 0.34 | 4.82 ± 0.41 |
| | July 2017 | 31.34 ± 0.65 | 4.7 ± 0.08 | 7.34 ± 0.2 | 39.25 ± 0.82 | 7.18 ± 0.06 | 0.18 ± 0.01 | 2.22 ± 0.07 | 0.65 ± 0.02 | 0.35 ± 0.07 | 20.18 ± 0.91 | 10.69 ± 0.41 | 10.15 ± 0.91 |
| After | August 2019 | 31.23 ± 0.22 | 4.79 ± 0.09 | 8.65 ± 0.23 | 68.12 ± 1.13 | 10.67 ± 0.26 | 0.07 ± 0.00 | 1.23 ± 0.02 | 0.51 ± 0.01 | 0.27 ± 0.05 | 12.29 ± 0.34 | 9.49 ± 1.00 | 7.59 ± 0.26 |
| | October 2019 | 20.54 ± 0.24 | 2.08 ± 0.1 | 8.36 ± 0.08 | 47.94 ± 4.58 | 10.17 ± 0.02 | 0.07 ± 0.00 | 1.48 ± 0.04 | 0.5 ± 0.01 | 0.3 ± 0.02 | 14.75 ± 0.25 | 10.68 ± 0.56 | 4.79 ± 0.24 |
| | January 2020 | 7.59 ± 0.3 | 1.52 ± 0.01 | 8.18 ± 0.14 | 33.85 ± 2.41 | 8.91 ± 0.15 | 0.04 ± 0.00 | 1.65 ± 0.1 | 0.48 ± 0.03 | 0.37 ± 0.06 | 17.26 ± 0.53 | 10.68 ± 0.43 | 2.33 ± 0.33 |
| | April 2020 | 19.03 ± 0.65 | 1.86 ± 0.04 | 8.24 ± 0.24 | 37.7 ± 1.72 | 8.92 ± 0.36 | 0.06 ± 0.01 | 1.38 ± 0.09 | 0.5 ± 0.03 | 0.34 ± 0.02 | 15.65 ± 0.11 | 10.21 ± 0.64 | 3.12 ± 0.51 |
| | July 2020 | 28.42 ± 0.1 | 4.71 ± 0.09 | 8.84 ± 0.13 | 70.43 ± 1.28 | 9.4 ± 0.37 | 0.07 ± 0.00 | 1.23 ± 0.09 | 0.5 ± 0.03 | 0.27 ± 0.01 | 13.19 ± 0.14 | 9.45 ± 0.37 | 6.54 ± 0.22 |
| | October 2020 | 22.77 ± 0.03 | 3.95 ± 0.04 | 8.04 ± 0.11 | 46.92 ± 1.29 | 7.91 ± 0.1 | 0.05 ± 0 | 1.13 ± 0.04 | 0.47 ± 0.02 | 0.25 ± 0.02 | 16.29 ± 0.41 | 10.25 ± 0.36 | 3.14 ± 0.13 |
| | April 2021 | 19.75 ± 0.04 | 1.45 ± 0.04 | 8.05 ± 0.23 | 37.72 ± 2.04 | 7.83 ± 0.1 | 0.06 ± 0.01 | 1.03 ± 0.03 | 0.39 ± 0.03 | 0.2 ± 0.02 | 17.03 ± 0.45 | 10.24 ± 0.34 | 2.12 ± 0.33 |
| | July 2021 | 28.53 ± 0.33 | 4.4 ± 0.04 | 8.07 ± 0.08 | 71.19 ± 2.04 | 9.46 ± 0.1 | 0.07 ± 0 | 1.15 ± 0.04 | 0.43 ± 0.02 | 0.24 ± 0.04 | 12.19 ± 0.34 | 9.36 ± 0.41 | 7.3 ± 0.24 |

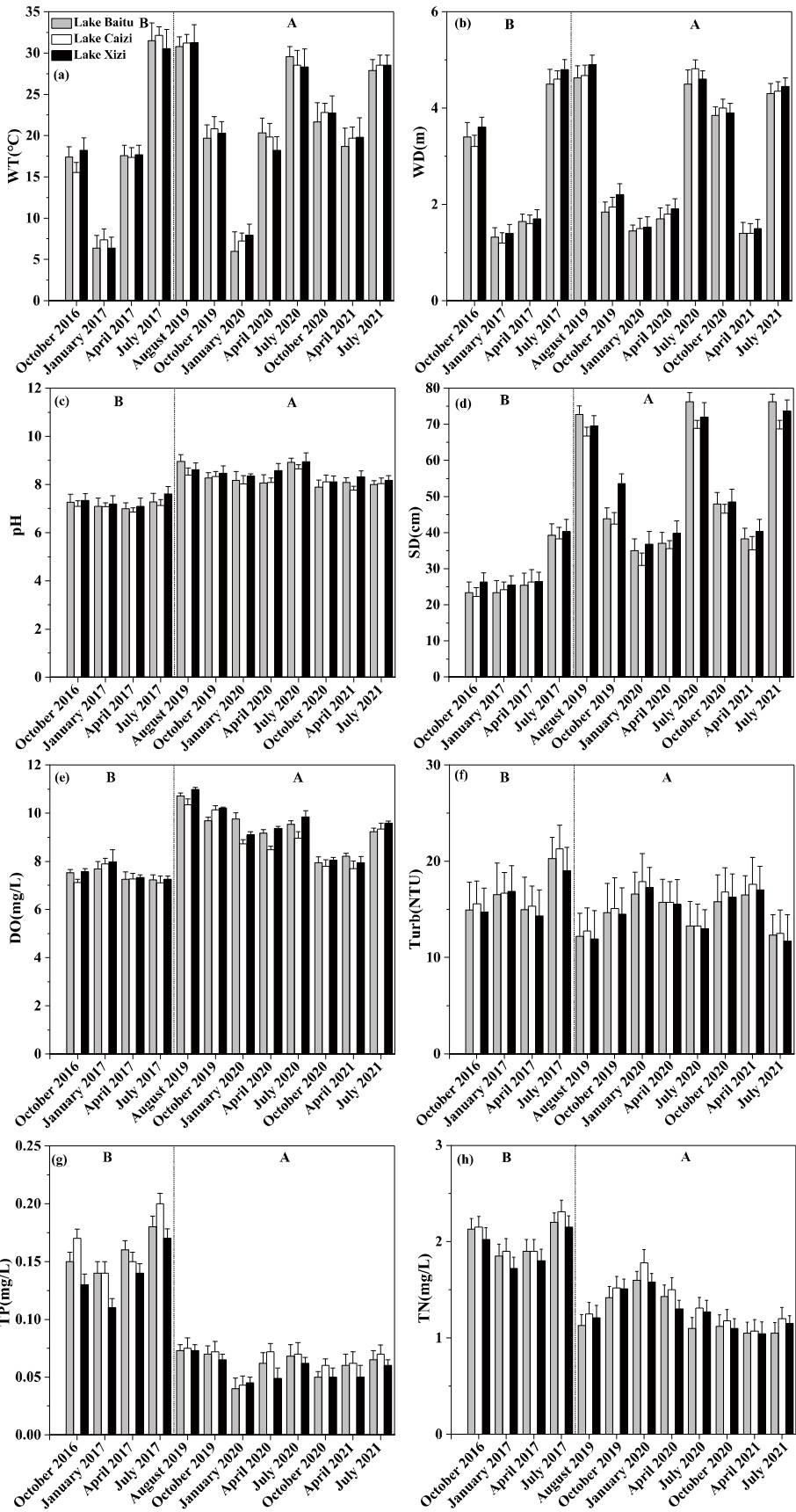

**Figure 2.** *Cont.*

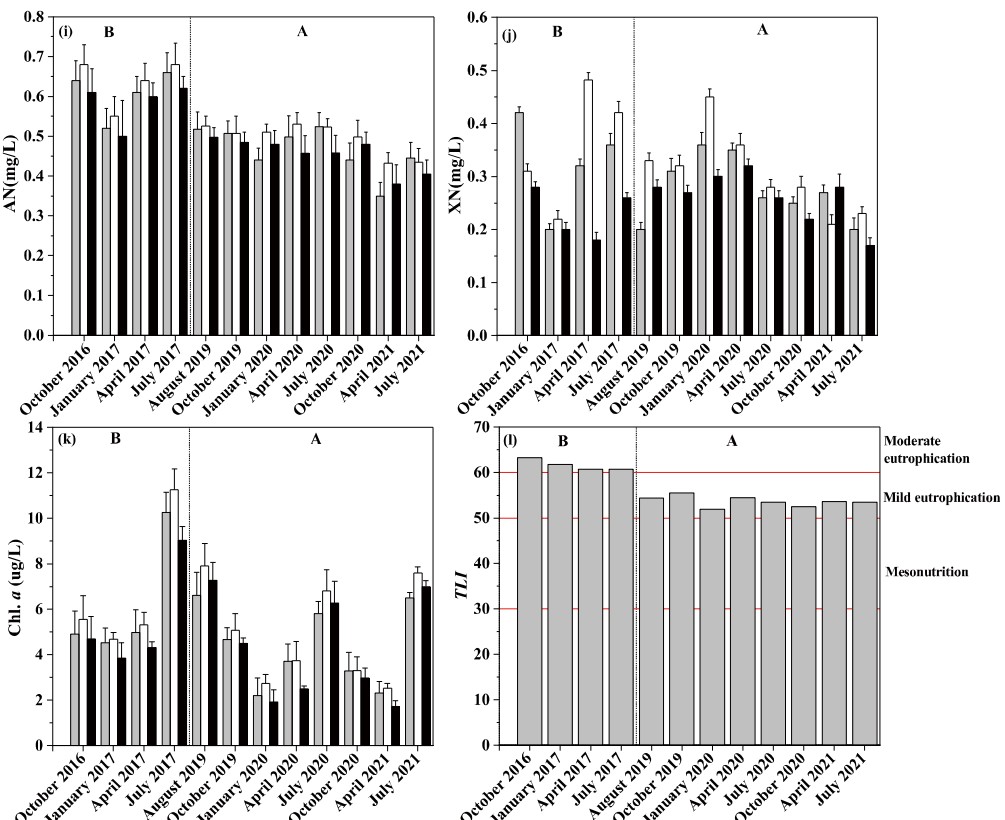

**Figure 2.** Spatial variation of different physical and chemical indices. Abbreviations was used in the diagram: (**a**) WT: water temperature, (**b**) WD: water depth, (**c**) pH, (**d**) SD: Secchi depth, (**e**) DO: dissolved oxygen, (**f**)Turb: turbidity, (**g**) TP: total phosphorus, (**h**) TN: total nitrogen, (**i**) AN: ammonia nitrogen, (**j**) XN: nitrate nitrogen, (**k**) Chl. *a*: chlorophyll *a*, and (**l**) *TLI*: comprehensive nutrient status index. The capital letters B and A represent before and after the restoration of aquatic vegetation, respectively. Vertical lines separated the before (October 2016 to July 2017) and after (August 2019 to July 2021) restoration periods. The red lines distinguished nutrient levels, with *TLI* between 30 and 40 being mesonutrition, between 50 and 60 being mild eutrophication, and between 60 and 70 being moderate eutrophication.

In terms of spatial variation, before the aquatic vegetation restoration, the pH, DO and SD of Xizi Lake were generally higher than those of Baitu Lake and Caizi Lake, but not significant. After the restoration, pH and DO also demonstrated the same spatial law (Figure 2c,e). However, the SD of Lake Baitu in summer was higher than that of Lake Caizi and Lake Xizi, by 9.23% and 4.39%, respectively, corresponding to values of $75.02 \pm 5.25$ cm, $68.09 \pm 4.32$ cm and $71.73 \pm 4.26$ cm, respectively (Figure 2d). Before the restoration of the aquatic vegetation, the concentration of Chl. *a* in Lake Baitu and Lake Caizi surpassed that of Lake Xizi in most cases. After the restoration of the aquatic vegetation, the concentration of Chl. *a* in Lake Baitu in the summer was 15.20% and 7.94% lower than what was in Lake Caizi and Lake Xizi, respectively, with corresponding values of 6.30 ug/L, 7.43 ug/L and 6.85 ug/L, respectively (Figure 2k). In terms of nutrient content, after the aquatic vegetation restoration, the TN content in Lake Baitu in the summer was significantly lower than that in Lake Caizi and Lake Xizi ($p < 0.05$) (Figure 2h).

### 3.2. Distribution of Aquatic Vegetation

We investigated the aquatic vegetation in the Caizi Lakes in the summer when the aquatic vegetation grew most vigorously. Before the restoration of the aquatic vegetation, we investigated eight main aquatic vegetation groups, which mainly grew in long strips along the lakeshore and at the entrance of river ditch, with only a few species and low

coverage, overlaying an area of 8.5 km². The areas of Lake Baitu, Lake Caizi and Lake Xizi were 7.4 km², 0.3 km² and 0.8 km², respectively. The main dominant groups of the lake were floating plants Ass. *E. crassipes*, Ass. *T. bispinosa* and emergent plants Ass. *Phragmites australis.* The main species included *Z. latifolia*, *N. peltatum*, *Polygonum lapathifolium*, *Potamogeton wrightii* and *Ceratophyllum demersum* (Figure 3a).

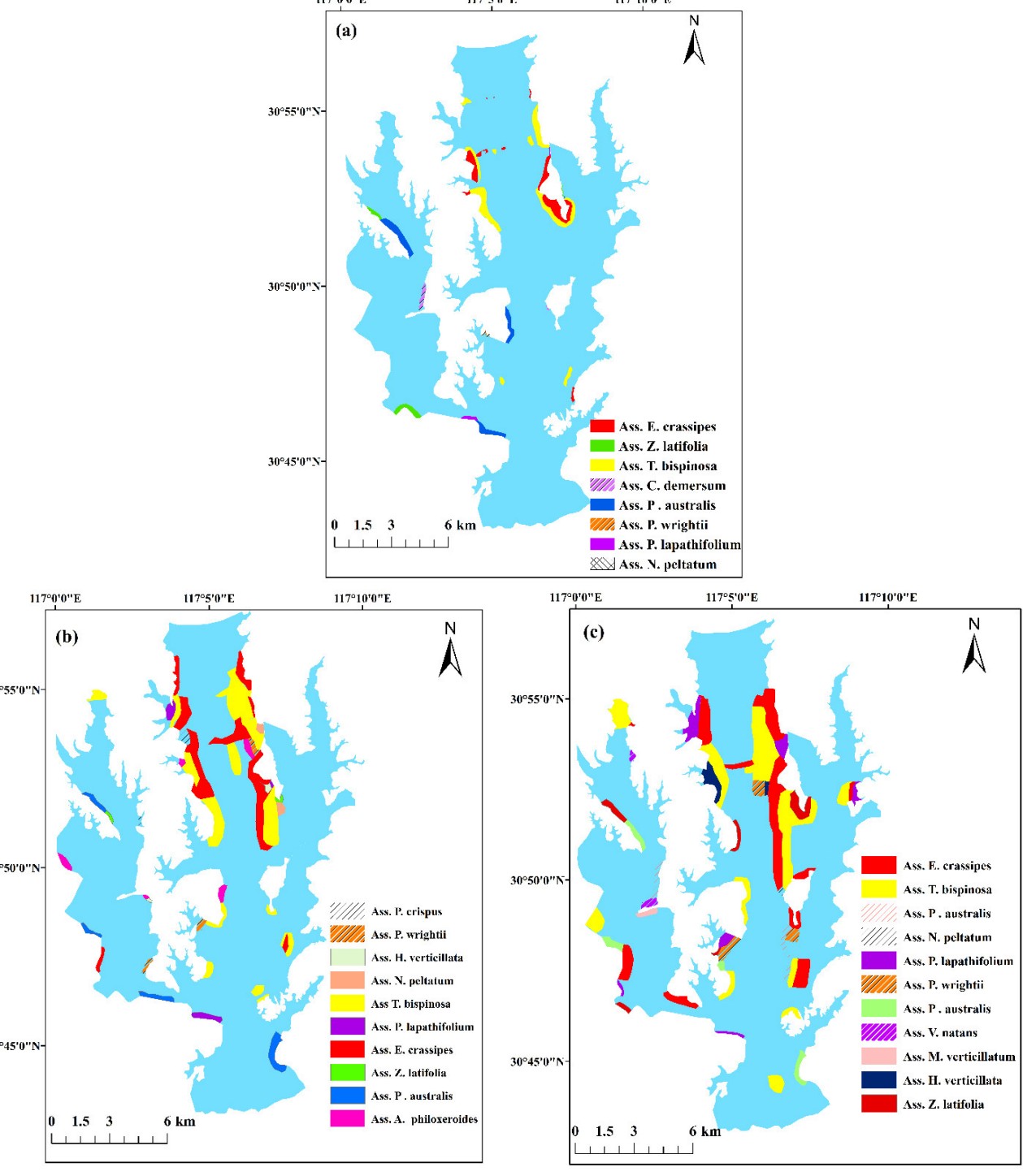

**Figure 3.** Distribution of aquatic vegetation in Caizi Lakes before and after removal of culture purse seine. Panel (**a**) presented the aquatic vegetation in July 2017 before restoration, and panels (**b**) and (**c**) presented the aquatic vegetation in August 2019 and July 2021 after restoration, respectively.

After the restoration of the aquatic vegetation, the dominant groups and vegetation coverage area of the lake expanded from 8.5 km² to 58.5 km², among which the areas of Lake Baitu, Lake Caizi and Lake Xizi were 46.3 km², 7.0 km² and 5.2 km², respectively. Their cover increased by 16.06%, 2.77% and 1.82%, respectively. The results showed that the aquatic vegetation cover of Lake Baitu rose significantly after restoration ($p < 0.05$). Floating plants, including Ass. *E. crassipes* and Ass. *T. bispinosa*, were the main dominant groups that were widely distributed along the east and west coasts of Lake Baitu. The main associated species of the lake were *N. peltatum*, *P. lapathifolium*, *Z. latifolia*, *Z. P. australis.*, *Vallisneria natans*, *P. wrightii*, *P. crispus*, *Hydrilla verticillata* and *Myriophyllum verticillatum* (Figure 3b,c). Generally speaking, the floating plants, mainly *E. crassipes* and *T. bispinosa*, recovered in the Lake Baitu after the purse seine was removed, and the coverage of submerged plants increased slightly, while the overlay of emergent plants did not change significantly ($p > 0.05$).

### 3.3. Temporal and Spatial Dynamics of Phytoplankton Cell Density, Biomass and Diversity Index

3.3.1. Temporal and Spatial Variation of Phytoplankton Cell Density

In terms of time, the cell density of phytoplankton changed obviously before and after the restoration of aquatic vegetation (Figure 4). Before the restoration of aquatic vegetation, the average cell density of phytoplankton in the Caizi Lakes was $21.04 \pm 4.57 \times 10^6$ cells/L. After restoration, it was $12.74 \pm 3.63 \times 10^6$ cells/L, which was significantly reduced by 39.45% ($p < 0.05$). This was mainly related to the reduction in cell density of Cyanobacteria. The average cell density of Cyanobacteria before and after recovery was $10.15 \times 10^6 \pm 4.06$ cells/L and $1.98 \times 10^6 \pm 1.00$ cells/L, respectively. After restoration, its relative abundance dropped significantly from 46.40% to 14.35% ($p < 0.01$). However, the relative abundance of Chlorophyta, Bacillariophyta and Euglenophyta climbed in different degrees after the restoration of aquatic vegetation. The relative abundance of Chlorophyta blossomed from 25.89% to 40.93% ($p < 0.01$), Bacillariophyta from 12.86% to 26.69% ($p < 0.01$) and Euglenophyta from 2.19% to 5.79% ($p < 0.01$). Seasonally, before the restoration of aquatic vegetation, the phytoplankton cell density changed significantly and showed the highest in summer and the lowest in winter, among which Cyanobacteria always occupied the main advantage, followed by Chlorophyta. After the restoration of aquatic vegetation, the cell density of phytoplankton was still the highest in summer and the lowest in winter, where the Chlorophyta dominated, followed by the Bacillariophyta (Figure 4).

From a spatial point of view, before the restoration of the aquatic vegetation, the phytoplankton cell density in Lake Caizi and Lake Baitu was notably higher than that in Lake Xizi ($p < 0.05$). After the restoration of the aquatic vegetation, there was no pronounced difference in the phytoplankton cell density between lakes ($p > 0.05$) (Figure 5).

3.3.2. Temporal and Spatial Variation of Phytoplankton Biomass

The difference in phytoplankton biomass before and after the aquatic vegetation restoration was obvious, which is $18.13 \pm 3.57$ mg/L and $9.72 \pm 2.55$ mg/L, respectively. The average biomass of phytoplankton was significantly reduced by 46.36% compared with that before restoration ($p < 0.05$), which was mainly attributed to the decrease in the Cyanobacteria (Figure 6). After the restoration, the relative abundance of Cyanobacteria biomass significantly shrank from $8.75 \pm 3.23$ mg/L to $1.26 \pm 0.58$ mg/L ($p < 0.01$). After the restoration of the aquatic vegetation, the relative abundance of Bacillariophyta and Chlorophyta increased by 19.73% ($p < 0.01$) and 10.00% ($p < 0.01$), respectively, and that of Euglenophyta and Cryptophyta climbed by 4.27% and 0.85%, respectively.

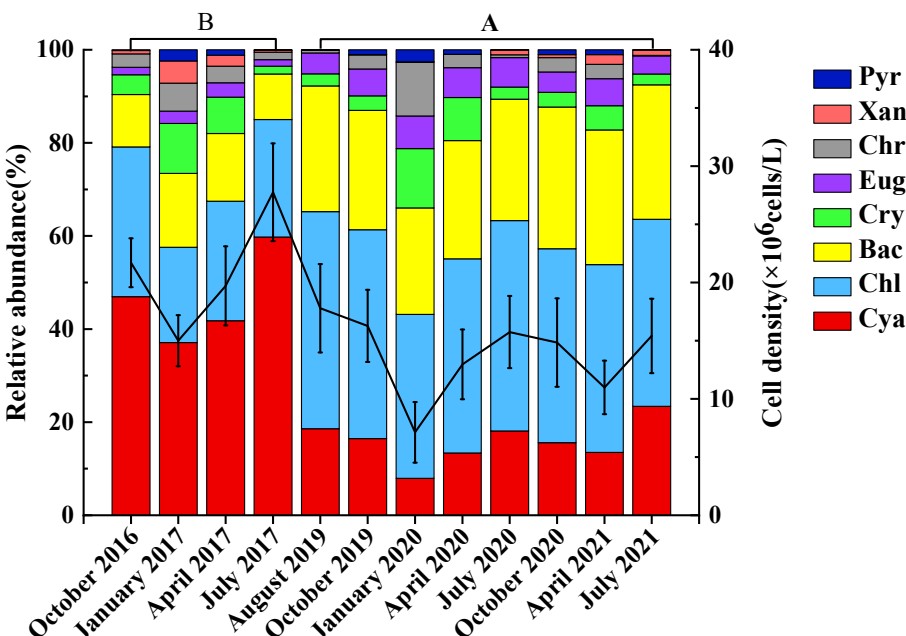

**Figure 4.** Temporal variation of phytoplankton cell density before and after aquatic vegetation restoration. The left axis or bar graph indicated the relative abundance of phytoplankton cell density, and the right axis or dark line indicated the average cell density (average and standard deviation) of all points at each sampling date in Caizi Lakes. Abbreviations was used in the diagram: Pyr: Pyrrophyta, Xan: Xanthophyta, Chr: Chrysophyta, Eug: Euglenophyta, Cry: Cryptophyta, Bac: Bacillariophyta, Chl: Chlorophyta, Cya: Cyanobacteria. Capital letters B and A indicated before and after aquatic plant restoration, respectively.

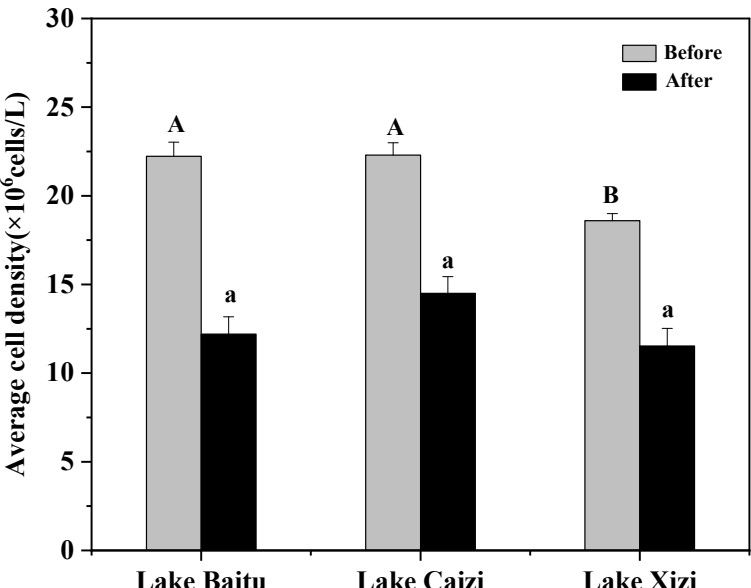

**Figure 5.** Spatial variation of phytoplankton cell density before and after the aquatic vegetation restoration. Upper case letters indicate significant differences between lakes before restoration, lower case letters indicate significant differences between lakes after restoration. The comparison between upper- and lower-case letters for each lake indicated significant differences before and after restoration. The same letter indicated a non-significant difference ($p > 0.05$), different letters indicated a significant difference ($p < 0.05$).

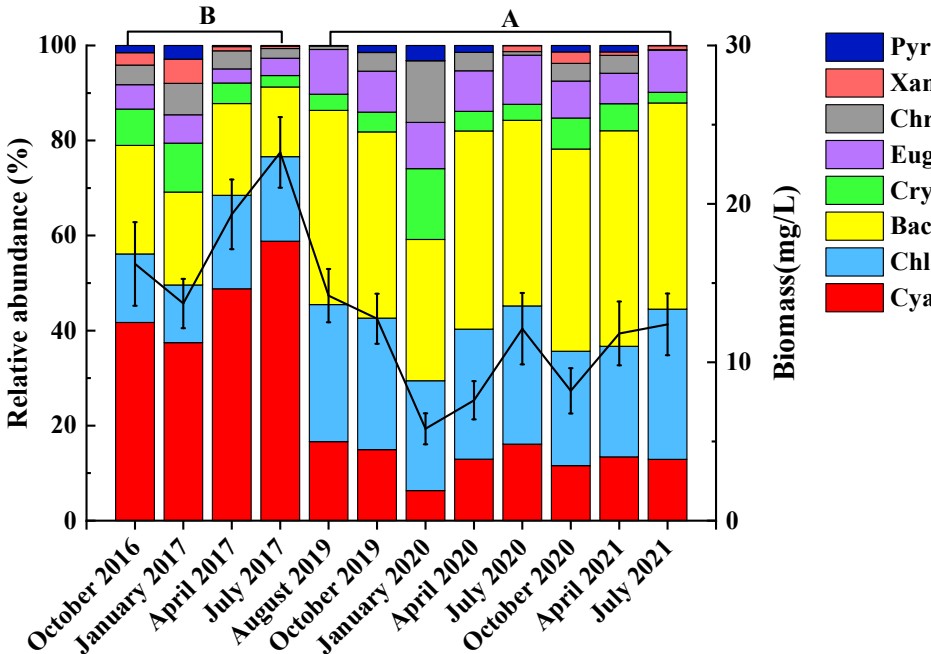

**Figure 6.** Temporal variation of the phytoplankton biomass before and after the aquatic vegetation restoration. The left axis or bar indicated the relative abundance of phytoplankton biomass, and the right axis or dark line indicates the mean biomass (average and standard deviation) of all points in Caizi Lakes on each sampling date. Abbreviations are used in the graph: Pyr: Pyrrophyta, Xan: Xanthophyta, Chr: Chrysophyta, Eug: Euglenophyta, Cry: Cryptophyta, Bac: Bac: Bacillariophyta, Chl: Chlorophyta, Cya: Cyanobacteria. Capital letters B and A indicated before and after aquatic plant restoration, respectively.

Seasonally, phytoplankton biomass showed obvious changes before and after the aquatic vegetation restoration (Figure 6). Before the restoration of the aquatic vegetation, the biomass of phytoplankton demonstrated a trend of high in the summer and spring and low in the autumn and winter, and Cyanobacteria and Bacillariophyta always dominated. After the restoration of the aquatic vegetation, the biomass was the highest in summer and the lowest in winter, and Bacillariophyta and Chlorophyta were always in a dominant position.

Spatially, the phytoplankton biomass of the Caizi Lakes showed significant spatial changes before and after the aquatic vegetation restoration, and their spatial change trends were inconsistent. However, the average biomass of Lake Caizi was always at the highest peak before and after the aquatic vegetation restoration (Figure 7). Before the aquatic vegetation restoration, the biomass of Lake Baitu and Lake Caizi was considerably higher than that of Lake Xizi ($p < 0.05$). After the aquatic vegetation restoration, the biomass of phytoplankton in Lake Caizi was significantly higher than that of Lake Baitu and Lake Xizi ($p < 0.05$). On the whole, the phytoplankton biomass of the three sub-lakes diminished after the restoration of the aquatic vegetation, especially in Lake Baitu, which decreased by 54.21% ($p < 0.01$), Lake Caizi and Lake Xizi, which dropped by 36.29% ($p < 0.01$) and 46.88% ($p < 0.01$), respectively.

To sum up, phytoplankton cell density and biomass were generally consistent in terms of spatial and temporal trends. Temporally, phytoplankton cell density and biomass were significantly reduced after the restoration of aquatic vegetation. The phytoplankton community was a Cyanobacteria-Chlorophyta-Bacillariophyta community before the restoration and a Chlorophyta-Bacillariophyta community after the restoration. In terms of seasonal variation, before and after the aquatic vegetation restoration, the cell density and biomass of the phytoplankton were the highest in the summer and the lowest in the winter. In terms of spatial variation, the cell density and biomass phytoplankton of Lake Baitu and Lake Caizi were appreciably higher than those of Lake Xizi before the aquatic vegetation

restoration. However, only the biomass of Lake Caizi was significantly higher than those of the other two lakes after the aquatic vegetation restoration.

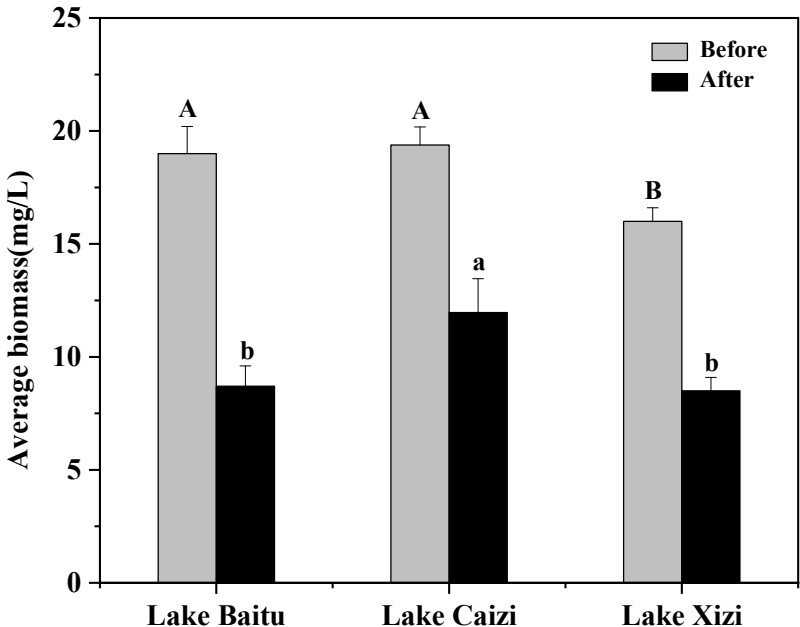

**Figure 7.** Spatial variation of the phytoplankton biomass before and after the aquatic vegetation restoration. Upper case letters indicate significant differences between lakes before restoration, lower case letters indicate significant differences between lakes after restoration. The comparison between upper- and lower-case letters for each lake indicated significant differences before and after restoration. The same letter indicated a non-significant difference ($p > 0.05$), different letters indicated a significant difference ($p < 0.05$).

### 3.3.3. Temporal and Spatial Variation of Phytoplankton Diversity

The results indicated that the Shannon–Wiener index ($H'$) increased significantly by 66.50% (from 1.81 to 3.01) after the aquatic vegetation restoration ($p < 0.05$). $H'$ was the highest in the winter before the restoration of the aquatic vegetation, and it was often the highest in the summer after the restoration of the aquatic vegetation. The Margalef index ($D$) rose by 46.20% (from 3.32 to 4.86) and the Pielou index ($J$) increased by 84.68% (from 0.36 to 0.67) after the aquatic vegetation restoration. There were pronounced differences in spatial changes in the phytoplankton diversity before and after the restoration of the culture purse seine (Figure 8). Before the restoration of the aquatic vegetation, $H'$, $D$ and $J$ in Lake Xizi were remarkably higher than those in Lake Caizi and Lake Baitu ($p < 0.05$). After the restoration of the aquatic vegetation, $H'$, $D$ and $J$ in Lake Xizi and Lake Baitu exceeded those in Lake Caizi in most cases. Among them, Lake Baitu $D$ was considerably higher than Lake Xizi and Lake Caizi in the summer after the aquatic vegetation restoration.

Generally speaking, the higher the diversity index, the more stable the phytoplankton community structure. In conclusion, $H'$, $D$ and $J$ indexes were significantly higher after aquatic vegetation restoration than before, so the phytoplankton community structure was more stable after the aquatic vegetation restoration. Combined with the diversity index to evaluate the water quality grading standards, it can be seen that before the restoration of aquatic vegetation, the Caizi Lakes were in a medium pollution state, and after the restoration they transformed into a light pollution state, in which the water quality of Lake Baitu improved obviously.

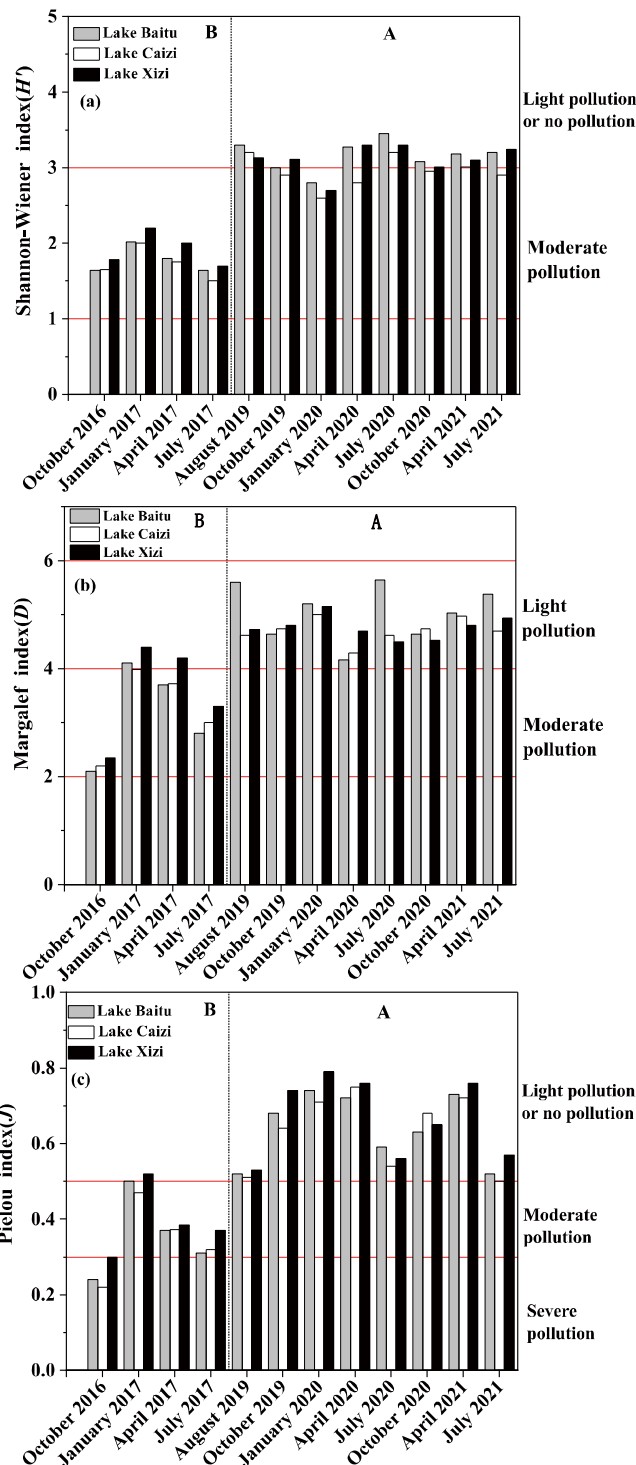

**Figure 8.** Temporal and spatial variation of the phytoplankton diversity index. Panels (**a–c**) repre-
sented the Shannon–Wiener diversity index of phytoplankton, the Margalef index and the Pielou
index, respectively. Capital letters B and A represented the before and after restoration of aquatic
vegetation, respectively. Vertical lines separated the period before (October 2016 to July 2017) and
after (August 2019 to July 2021) recovery. According to the diversity index, the nutrient level of lakes
was judged and distinguished by red lines. In Figure a, when *H* is between 1 and 3, it is moderate
pollution, and when *H* is greater than 3, it is light pollution or no pollution. In Figure b, when *D* is
between 2 and 4, it is moderate pollution, and when *D* is between 4 and 6, it is light pollution. Figure
c When *J* is between 0 and 0.3, it is severe pollution, between 0.3 and 0.5, it is moderate pollution, and
greater than 0.5, it is light pollution or no pollution.

### 3.4. Dominant Species of Phytoplankton

We identified 18 dominant phytoplankton species before and after the aquatic vegetation restoration. The dominant species were *Microcystis aeruginosa*, *Phormidium tenue*, *Oscillatoria tenuis*, *Dolicospermum circinale*, *Merismopedia minima*, *Aphanizomenon flos-aquae* and *Aphanocapsa elachista*. Secondly, there were four species of Chlorophyta (*Monorapidium contortum*, *Scenedesmus quadricauda*, *Chlorella vulgaris*, *Monactinus simplex*), four species of Bacillariophyta (*Aulacoseira granulata*, *Cyclotella meneghiniana*, *Ulnaria acus* and *Encyonema perpusillum*), and one species each of Cryptophyta, Euglenophyta and Chrysophyta, which were *Cryptomonas erosa*, *Trachelomonas superba*, and *Dinobryon bavaricum*, respectively (Table 2).

**Table 2.** Dominant species and temporal dynamics of phytoplankton.

| Phylum | Dominant Species | Dominance of Phytoplankton Species before Restoration (Y) | | | | Dominance of Phytoplankton Species after Restoration (Y) | | | |
|---|---|---|---|---|---|---|---|---|---|
| | | Spring | Summer | Autumn | Winter | Spring | Summer | Autumn | Winter |
| Cyanobacteria | M. aeruginosa | 0.06 | 0.19 | 0.18 | 0.05 | | 0.03 | 0.02 | |
| | P. tenue | 0.02 | 0.17 | 0.04 | | | | | |
| | O. tenuis | 0.22 | | | 0.11 | | 0.03 | 0.02 | |
| | D. circinale | 0.05 | | 0.13 | | | | 0.03 | |
| | A. flos-aquae | | 0.02 | 0.02 | | | | 0.02 | |
| | M. tenuissima | | 0.02 | 0.02 | | | | | |
| | A. elachista | | | | | 0.02 | | | |
| Chlorophyta | M. contortum | | | 0.04 | 0.03 | | | | 0.02 |
| | S. quadricauda | | | 0.03 | | | 0.12 | 0.03 | 0.06 |
| | C. vulgaris | 0.02 | 0.05 | 0.02 | | 0.02 | 0.05 | | 0.05 |
| | M. simplex | | | 0.02 | | 0.03 | | | 0.02 |
| Bacillariophyta | A. granulata | 0.02 | 0.12 | 0.03 | 0.16 | 0 | 0.02 | 0.02 | 0.03 |
| | U. acus | | | | | 0.02 | 0.04 | | 0.06 |
| | E. perpusillum | | | | | 0.02 | 0.02 | | 0.02 |
| | C. meneghiniana | | | | | | 0.02 | 0.02 | |
| Cryptophyta | C. erosa | | | | 0.02 | 0.03 | | | 0.04 |
| Euglenophyta | T. superba | | | | | 0.02 | | | |
| Chrysophyta | D. bavaricum | | | | | | | | 0.03 |

Before the restoration of aquatic vegetation, there were 12 dominant phytoplankton species in Caizi Lakes, which grew to 16 species after restoration. In addition, the dominant groups of phytoplankton also changed greatly. Cyanobacteria were dominant before the aquatic vegetation restoration, and the species with high dominance included *M. aeruginosa*, *P. tenue*, *A. variabilis* and *O. tenuis*. However, green algae and diatoms were consistently dominant after aquatic vegetation restoration. In general, Cyanobacteria, Chlorophyta and Bacillariophyta were the dominant phytoplankton species in the Caizi Lakes before the restoration of aquatic vegetation, and changed to Chlorophyta and Bacillariophyta after the restoration.

### 3.5. Relationship between Phytoplankton and Environmental Factors

The results of the Pearson correlation analysis showed that phytoplankton cell density was highly significantly positively correlated with WT, WD, TN, TP, AN, XN and Chl. *a* ($p < 0.01$), significantly positively correlated with SD ($p < 0.05$); biomass was highly significantly positively correlated with WT, SD, TP and Chl. *a* ($p < 0.01$), significantly positively correlated with AN and XN ($p < 0.05$), and significantly negatively correlated with N/P ($p < 0.05$). After the restoration of the aquatic vegetation, the cell density of the phytoplankton was highly significantly positively correlated with WT and TP ($p < 0.01$), significantly positively correlated with WD, SD and Chl. *a* ($p < 0.05$), and significantly negatively correlated with Turb ($p < 0.05$); biomass was highly significantly positively correlated with TP ($p < 0.01$), WT and SD ($p < 0.05$), and significantly negatively correlated with Turb ($p < 0.05$) (Table 3).

**Table 3.** Relationship between the cell density and biomass of the phytoplankton and environmental factors before and after the aquatic vegetation restoration.

| Parameter | Before Aquatic Vegetation Restoration | | After Aquatic Vegetation Restoration | |
|---|---|---|---|---|
| | Cell Density | Biomass | Cell Density | Biomass |
| | r | r | r | r |
| WT | 0.965 ** | 0.936 ** | 0.934 ** | 0.809 * |
| WD | 0.944 ** | 0.697 | 0.723 * | 0.497 |
| SD | 0.801 * | 0.860 ** | 0.751 * | 0.715 * |
| pH | 0.408 | 0.187 | 0.439 | 0.436 |
| DO | −0.484 | −0.583 | 0.387 | 0.423 |
| TP | 0.961 ** | 0.923 ** | 0.891 ** | 0.879 ** |
| TN | 0.874 ** | 0.614 | 0.171 | 0.434 |
| AN | 0.942 ** | 0.769 * | 0.327 | −0.064 |
| XN | 0.884 ** | 0.786 * | −0.426 | −0.678 |
| Turb | 0.543 | 0.54 | −0.740 * | −0.713 * |
| Chl. *a* | 0.890 ** | 0.871 ** | 0.795 * | 0.706 |
| N/P | −0.496 | −0.780 * | −0.707 | −0.466 |

Note: * represents a significant correlation ($p < 0.05$), and ** represents a very significant correlation ($p < 0.01$).

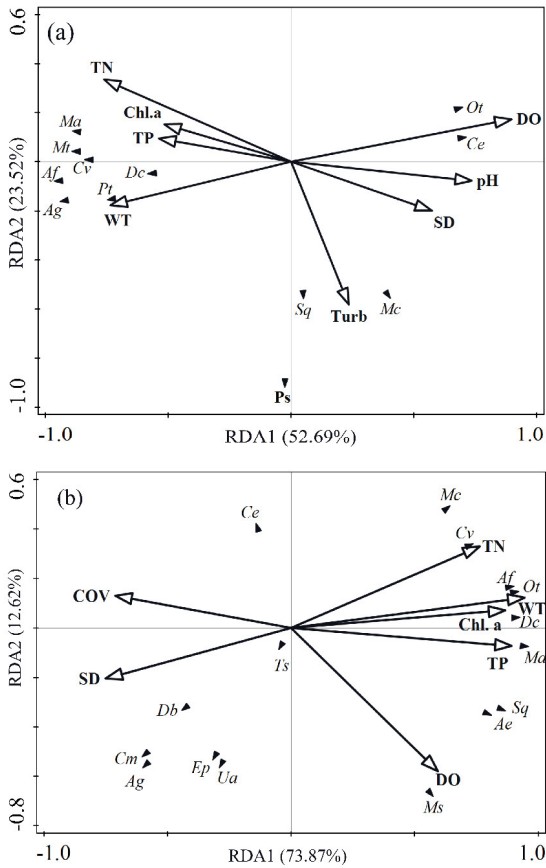

**Figure 9.** Redundancy analysis of the biomass of the dominant phytoplankton species (solid arrow) and the environmental factors (hollow arrow) before (**a**) and after (**b**) the aquatic vegetation restoration. Abbreviated as, Ma: *M. aeruginosa*; Pt: *P. tenue*; Ot: *O. tenuis*; Dc: *D. circinale*; Mt: *M. tenuissima*; Af: *A. flos-aquae*; Ae: *A. elachista*; Cv: *C. vulgaris*; Mc: *M. contortum*; Sq: *S. quadricauda*; Ms: *M. simplex*; Ag: *A. granulata*; Ua: *U. acus*; Ep: *E. perpusillum*; Cm: *C. meneghiniana*; Ce: *C. erosa*; Ts: *T. superba*; Db: *D. bavaricum*. WT: water temperature, SD: Secchi depth, DO: dissolved oxygen, Turb: turbidity, TN: total nitrogen, TP: total phosphorus, pH, Chl. *a*: chlorophyll *a*, COV: cover of aquatic vegetation.

The RDA results showed that all selected parameters on the first two axes explained 76.21% and 86.49% of the changes in the phytoplankton community before and after the restoration of aquatic vegetation, respectively, which better explained the relationship between species and environmental factors. The results showed that WT, TN, DO and Turb were the main environmental parameters affecting the dominant species of phytoplankton before the purse seine was removed. As can be seen from Figure 9a, *M. aeruginosa*, *M. tenuissima*, *D. circinale*, *A. elachista*, *A. flos-aquae*, *P. tenue* and *A. granulata* were positively correlated with WT, TN, TP and Chl. *a*, and negatively correlated with DO; SD and pH. *O. tenuis* and *C. erosa* were positively correlated with DO and negatively correlated with WT. *S. quadricauda*, *M. simplex* and *M. contortum* were positively correlated with Turb. WT, TP, TN, DO and SD were the main factors affecting the phytoplankton community structure after the purse seine removal. As illustrated in Figure 9b, *A. flos-aquae*, *O. tenuis*, *D. circinale* and *M. aeruginosa* were correlated with WT, TP and Chl. *a*, and negatively correlated with pH and SD. *M. contortum* and *C. vulgaris* were negatively correlated with SD, and *S. quadricauda*, *M. simplex* and *M. tenuissima* were positively correlated with DO and negatively correlated with pH. *C. meneghiniana*, *A. granulata*, *U. acus*, *E. perpusillum*, *D. bavaricum* and *T. superba* were positively correlated with SD and negatively correlated with TN, WT and TP. The main environmental factors affecting the dominant phytoplankton species changed after aquatic vegetation restoration, indicating that aquatic vegetation restoration provided the driving force for phytoplankton succession.

## 4. Discussion

### 4.1. Restoration of the Aquatic Vegetation and Its Impact on the Water Environment

After the purse seine was removed, the species diversity of the aquatic vegetation grew, the area of aquatic vegetation recovered from 8.5 km$^2$ to 58.5 km$^2$, and the coverage increased by 20.6%. This was consistent with the previous research results. Some scholars showed that the removal of large-scale aquaculture purse seine can promote the improvement of aquatic vegetation diversity [26]. Others proved that the aquatic vegetation in the Huayang Lakes recovered rapidly after dismantling the purse seine [6]. This may be because the density of herbivorous fish declined, which reduced the damage to seedlings and fruits of the aquatic vegetation and increased the possibility of the germination and growth of aquatic vegetation [27]. In addition, the bait needed for aquaculture was dissolved in water, leading to a decrease in the transparency of the water body, which was not conducive to the growth of the aquatic vegetation. After the removal of the seine, the reduction in human disturbance allowed the lake ecosystem to gradually recover in its natural state and promoted the recovery of aquatic vegetation.

We found that the restoration of the aquatic vegetation in different sub-lakes was different. The aquatic vegetation cover of Lake Baitu, Lake Caizi and Lake Xizi increased by 19.11%, 2.77% and 1.82%, respectively, which determined that Lake Baitu had the best aquatic vegetation restoration effect. Floating plants, mainly *E. crassipes* and *T. bispinosa* were widely distributed in the east and west coasts of Lake Baitu. Compared to submerged plants, the larger leaf surface floating on the water surface can obtain light directly and this determined that the submerged plants were one of the vegetation types with reproductive ability [12]. In addition, the higher number of floating plants before the recovery of aquatic plants provided the basis for their subsequent sexual and nutritional reproduction in Lake Baitu, which all led to the rapid recovery of floating vegetation in Lake Baitu.

According to the classification standard of lake eutrophication, our results showed that, after the restoration of the aquatic vegetation, the *TLI* index was 53.42, representing a 13.23% reduction, and the overall water quality changed from moderate eutrophication to mild eutrophication. The significant positive correlation between the aquatic vegetation and SD and the significant negative correlation between the aquatic vegetation and TN and TP indicated (Figure 9b) that the restoration of the aquatic vegetation was the main reason for the improvement of the water quality in the Caizi Lakes. It is well known that aquatic vegetation has the positive feedback effect of stabilizing sediments, alleviating eutrophica-

tion and improving water quality [28–30] and can also absorb nutrients such as nitrogen and phosphorus in water column and sediments through the leaves and roots system [31]. In this study, SD and DO increased by 67.64% and 22.74%, especially in the summer, while SD rose by 32.37 cm. Some scholars showed that the presence of macrophytes facilitated the maintenance of high transparency in the water column [32]. In addition, aquatic vegetation effectively absorbed carbon dioxide and released oxygen during photosynthesis, thus improving the DO concentration in the Caizi Lakes. After the restoration of the aquatic vegetation, water column TN and TP concentrations were reduced by 33.00% and 22.53%, respectively, especially in the summer when the aquatic vegetation grew most vigorously, the efficiency of improving the water transparency and removing the total nitrogen in Baitu Lake was appreciably higher than that in Lake Caizi and Lake Xizi (Figure 2). Aquatic vegetation can improve the permeation of dissolved oxygen, change the microorganisms related to the nitrogen cycle at the sediment-water interface, and couple nitrification and denitrification, thus improving the nitrogen removal capacity [33]. The rhizosphere of the aquatic vegetation secretes organic substances. This can affect denitrifying bacteria and ammonifying bacteria in sediments and promote the nitrogen cycle [34], all of which may have contributed to the significantly lower TN concentration in Lake Baitu than in Lake Caizi and Lake Xizi in summer. To sum up, the improvement of water quality in the Caizi Lakes was closely related to the restoration of the aquatic vegetation.

### 4.2. Effects of Aquatic Vegetation Restoration on the Cell Density, Biomass and Diversity of Phytoplankton

Our research demonstrated that aquatic vegetation restoration significantly reduced the cell density and biomass of phytoplankton. As mentioned above, the concentrations of TP and TN in water significantly fell after aquatic vegetation restoration. Combined with the fact that phytoplankton was positively correlated with TN and TP concentrations before aquatic plant restoration, however, after aquatic plant restoration, phytoplankton was positively correlated with TP and poorly correlated with TN; it was hypothesized that TP was the main factor limiting the growth of phytoplankton. As other studies have demonstrated, TP is the key factor limiting in phytoplankton growth [35]. Phytoplankton reproduction was limited by the reduction in nutrient concentration, which was closely related to the recovery of aquatic plants.

According to the Pearson correlation analysis (Table 3), we found that aquatic plant cover was significantly and negatively correlated with phytoplankton cell density and biomass, which was not only related to the fact that aquatic plants can reduce nitrogen and phosphorus nutrients in the water column through physical feedback mechanisms, but may also be caused by the biofeedback mechanism between aquatic vegetation and phytoplankton. Studies have shown that phytoplankton cell density and biomass tend to be lower in the presence of aquatic vegetation [6]. In areas where *E. crassipes* and *T. bispinosa* are present, water transparency increases significantly and phytoplankton cell density decreases significantly [36,37]. The inhibition mechanism of the aquatic vegetation on phytoplankton growth is complex. On the one hand, aquatic vegetation can directly inhibit phytoplankton growth through allelopathy and competition. Studies have established that *E. crassipes* inhibit algal growth by secreting allelochemicals that alter the cell structure of algae and reduce their photosynthesis [38]. The aquatic vegetation, mainly *E. crassipes* and *T. bispinosa*, may inhibit phytoplankton growth by secreting allelochemicals after the removal of seine culture. Moreover, aquatic vegetation also inhibits phytoplankton growth by competing for light and nutrients [11]. On the other hand, indirect effects such as predation of aquatic organisms may also affect the phytoplankton. Aquatic vegetation provides shelter for zooplankton to resist the predation of fish, and zooplankton controls the density of phytoplankton through the top-down predation effect [39]. In regard to all the above-mentioned mechanisms, the reduction in phytoplankton was related to the restoration of the aquatic vegetation after the removal of the purse seine.

The diversity results showed that phytoplankton diversity increased noticeably after the purse seine was removed. This was consistent with many previous studies that aquatic vegetation can promote phytoplankton diversity [6,40]. After the removal of the seine, the Shannon–Wiener index and richness index were obviously higher in Lake Baitu, which had the best aquatic vegetation restoration, than in other lakes, which was related to the fact that the restoration of various aquatic vegetation provided different habitats for phytoplankton. Different aquatic vegetation has specific allelopathy on phytoplankton. It has been suggested that the increase in phytoplankton diversity after aquatic vegetation restoration was related to the chemosensory effects of aquatic vegetation [6].

### 4.3. Effects of Aquatic Vegetation Restoration on Phytoplankton Species Change

Our research confirmed that the restoration of the aquatic vegetation led to significant changes in the species composition of phytoplankton communities. The phytoplankton changed from a Cyanobacteria-Chlorophyta-Bacillariophyta dominant community to a Chlorophyta-Bacillariophyta dominant community after aquatic vegetation restoration, and the phytoplankton density and biomass decreased significantly. This was mainly attributed to the significant change in Cyanobacteria, which was 80.49% lower than before the aquatic plant restoration, and the relative abundance of Chlorophyta, Bacillariophyta and Cryptophyta showed different degrees of increase. Among them, the Bacillariophyta dominant species changed greatly after aquatic plant restoration, from being dominated by *Centricae* to *Pennatae*. This was in general agreement with the results of others, where aquatic vegetation restoration significantly reduced the relative abundance of Cyanobacteria and rose the relative abundance of Chlorophyta, Bacillariophyta and Cryptophyta in the Huayang lakes [6].

In this study, the aquatic vegetation restoration significantly reduced the dominance of harmful Cyanobacteria (*M. aeruginosa*, *O. tenuis*, *A. flos-aquae*, *D. circinale*). Combined with an RDA analysis (Figure 9b), most of the harmful Cyanobacteria were positively correlated with TN and TP, and negatively correlated with aquatic vegetation coverage, so it was judged that the decrease in Cyanobacteria was related to nutrient reduction and aquatic vegetation restoration. Studies on the reduction in Cyanobacteria caused by aquatic vegetation restoration have been reported many times. Some scholars have found that Cyanobacteria biomass in vegetated and non-vegetated areas differed significantly, with relative abundances of 18.96% and 34.05%, respectively [41]. Laboratory results confirmed that *E. crassipes* could obviously inhibit the growth of *M. aeruginosa* and promote the decay of algae cells [42,43].

This study showed that aquatic plant restoration not only increased Bacillariophyta abundance, but also changed species significantly, from Centricae (*A. granulata*) to Pennatae (*U. acus*, *E. perpusillum*). *U. acus* and *E. perpusillum* are usually considered as attached phytoplankton, and the restoration of aquatic vegetation provides the conditions for the survival and reproduction of attached algae [44]. RDA analysis showed that *U. acus* and *E. perpusillum* were negatively correlated with TN and TP. Therefore, the decrease in nutrient concentration in the water column after aquatic vegetation restoration was also responsible for the increase in the relative abundance of attached Bacillariophyta. As other studies have shown, *U. acus* and *E. perpusillum* are generally considered to live in cleaner water [45]. The rise in relative abundance of Chlorophyta was also the main feature of phytoplankton community succession. Chlorophyta cells are usually very small (1–5 µm), and small cells have a high surface area-to-volume ratio, which promotes a higher nutrient acquisition rate. This feature may contribute to the rapid reproduction of Chlorophyta after aquatic vegetation restoration. Moreover, after the restoration of aquatic vegetation, the number of Cyanobacteria decreased, which may have released the competitive pressure on Chlorophyta, allowing it to grow.

A common discovery about shallow lakes is that the relative abundance of small and usually flagellated phytoplankton increases in the presence of aquatic vegetation. Some scholars have recorded that the species composition has changed from Cyanobacteria to

Cryptophyta in lakes from no vegetation to dense vegetation [46]. RDA analysis showed that *C. erosa* was positively correlated with aquatic vegetation coverage (Figure 9b). Many factors may contribute to the survival of Cryptophyta under aquatic vegetation. Firstly, compared with non-motile species, the flagellate's initiative enables it to utilize nutrient sources in different spaces [47]; finally, most flagellates are facultative heterotrophic and can absorb dissolved organic carbon from the secretions of the aquatic vegetation. Cryptophyta, as a typical phytoplankton group with flagella, prefer a still water environment [45]. Its increase after the recovery of aquatic vegetation could be that the new habitat form increased its survival advantage.

Generally speaking, the decrease in Cyanobacteria, the increase in Chlorophyta, Bacillariophyta and Cryptophyta, and the change in Bacillariophyta species were the main reasons for the change in the phytoplankton community pattern in the Caizi Lakes. This was closely related to the restoration of aquatic vegetation, which can affect the phytoplankton community, not only directly through a biofeedback mechanism but also indirectly by changing the environmental characteristics of the water column, including an increase in DO and SD and a decrease in TN and TP. These results confirmed our hypothesis that the restoration of aquatic vegetation significantly affected the composition of the phytoplankton community after the removal of the seine.

### 4.4. Mechanism of Formation of Alternative States of the Lake

Studies have shown that two different alternative equilibria states exist in shallow lakes, namely a clear state dominated by submerged vegetation and a turbid state dominated by phytoplankton [48]. Ecosystems respond to external changes and perturbations in different ways and may switch from one state to another when external condition thresholds are exceeded, but it can change back to the original state at different thresholds when conditions are reversed [49].

The Caizi Lakes are typical shallow lakes that have undergone different alternative states over the years. Under their original conditions, the Caizi Lakes had clear water quality and abundant submerged vegetation. However, with the rapid development of aquaculture in the Caizi Lakes, the water quality has greatly reduced, resulting in increased turbidity and reduced transparency of the water column. This has caused a catastrophic change in the lakes to a turbid state, dominated by phytoplankton due to increased turbidity—through various mechanisms—and loss of vegetation, which was difficult to recover naturally [50].

After the large-scale removal of farming seines, the Caizi Lakes changed from a turbid state dominated by phytoplankton to a clear state dominated by aquatic vegetation, and these changes may be attributed to the positive feedback between water transparency and aquatic vegetation. The positive effect of aquatic vegetation, which can improve environmental conditions and improve water clarity for better growth, may help create positive feedbacks. However, this positive feedback effect was more pronounced for submerged vegetation, which has more stringent transparency requirements for growth. Our study showed that after the removal of the seine net, the aquatic vegetation in the Caizi Lakes was dominated by floating vegetation rather than submerged vegetation, which is an alternative steady state occurrence. This was significant in that it indicated that there was a hysteresis in the lakes during the alternative state transition. It has been shown that a return to a state dominated by pristine, clear and submerged vegetation requires a reduction in nutrient loading to levels well below those that existed when the vegetation collapsed [51]. In addition, increased aquatic vegetation can provide a refuge for zooplankton. This increases the risk of phytoplankton predation by zooplankton and also reduces the competitive effects of phytoplankton and aquatic vegetation, thus increasing the growth advantage of aquatic vegetation. In summary, after the removal of the seine, the Caizi Lakes changed from a turbid state dominated by phytoplankton to a clear state dominated by aquatic vegetation, and this change may be related to the positive feedback between aquatic vegetation and water transparency, and may also be caused by the interaction between aquatic vegetation and aquatic organisms.

## 5. Conclusions

In this study, we explored the response of phytoplankton to aquatic vegetation recovery after the removal of purse seine. The results showed that aquatic vegetation recovered in the Caizi Lakes after the removal of farming seine nets, which improved lake water quality and drove the evolution of phytoplankton community structure. Our results were consistent with the performance of some other shallow lakes such as the Huayang Lakes. According to the theory of alternative stable states in shallow lakes, this state of transformation from turbid water to clear water in the Caizi Lakes after the removal of the seine was predicted. When the anthropogenic disturbance was removed, the lake naturally recovered and tended to a stable structure. Therefore, it is recommended that the lake seine be completely removed to reduce anthropogenic disturbance in the lake. This study was conducted with a view to providing a basic theoretical basis for evaluating the restoration effect of aquatic vegetation from the perspective of phytoplankton, which was indicative of lake protection and ecological restoration. However, whether the role of aquatic vegetation on aquatic environmental restoration is consistent in different environments such as the ocean or reservoirs is still unknown and needs further analysis. In addition, when we explored the effect of seine removal on phytoplankton community structure, only the biological factor aquatic vegetation was selected, which can be further explored in follow-up work in combination with bird, fish and zooplankton changes.

**Author Contributions:** Conceptualization: Z.Z.; Data curation: W.Z., Z.L. and W.G.; Investigation: W.Z., Z.L. and W.G.; Methodology: Z.Z.; Writing—original draft: W.Z.; Writing—review and editing: W.Z. and Z.Z.; Funding acquisition: Z.Z. All authors have read and agreed to the published version of the manuscript.

**Funding:** This research was supported by Joint Research Project for the Yangtze River Conservation (Phase I), China (No. 2019-LHYJ-01-0212, 2019-LHYJ-01-0212-17).

**Institutional Review Board Statement:** Not applicable.

**Informed Consent Statement:** Not applicable.

**Data Availability Statement:** The data presented in this study are available on request from the corresponding author Zhongze Zhou. E-mail: zhzz@ahu.edu.cn.

**Acknowledgments:** We would like to thank Zhenzhong Liu, Wenli Guo for assistance with data collection. We would also like to thank Zhongze Zhou for his suggestions on the thesis. We would like to thank Marci Baun from the University of California, Los Angeles, for editing the paper.

**Conflicts of Interest:** The authors declare no conflict of interest.

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
