# Peer review of "Effect of Aquatic Vegetation Restoration after Removal of Culture Purse Seine on Phytoplankton Community Structure in Caizi Lakes"

_diversity, doi:10.3390/d14050395_

Round 1

Reviewer 1 Report

Review- Manuscript ID: diversity-1671136

Effect of aquatic vegetation restoration after removal of culture purse seine on phytoplankton community structure in Caizi Lake

In the manuscript entitled “Effect of aquatic vegetation restoration after removal of culture purse seine on phytoplankton community structure in Caizi Lake”, the authors detailed an investigation documenting water quality parameters, aquatic macrophytes distribution and phytoplankton community biomass, composition, diversity, richness and evenness before and after the removal of purse seine in a Chinese lake composed of three sub-lakes, which is directly connected to the middle and lower reaches of the Yangtze River. Globally, this manuscript addresses the effects of the restoration of the lake’s aquatic vegetation from an ecologically very-interesting phytoplankton point of view, which is often under-studied compared to more basic analyses of water quality and macrophytes coverage studies. Using 12 samplings spanning over almost five years and at 20 different stations within the lake, the authors reported important changes in phytoplankton biomass and community composition which they related to changes in water quality and macrophytes coverage induced by the purse seine removal. The study was well-designed and enabled the detection of the consequences of the removal of purse seine on phytoplankton. However, I found that both the Materials and Methods and the Results sections need heavy revisions as many parts are unclear and/or incomplete. In addition, most figure captions are not precise enough and do not present sufficient information to fully understand the figures they are referring to. The Introduction and Discussion sections seem quite well-structured, although some paragraphs could be shortened and better-structured. Nonetheless, the Discussion section relates well the reported results with other studies. The Conclusion section sums up well the major findings of the study, however I think it is too descriptive, and more contextualization and generalization should be done. Finally, many small errors greatly reduce the overall quality of the manuscript. In this regard, I strongly recommend to use a professional language revision service. I have numerous specific comments that are listed below. 

Abstract

L15, 16, 17, 18: I think it is necessary to specify that the values presented represent average of all the stations and all the sampling dates before and after the restoration.

Introduction

L28: Please replace “are” by “is”.

L31-34. I think that this sentence could be spited to 2 sentences, that would make it easier to read.

L35. I would change “deposit” for “settle”.

L37. This sentence should be reformulated as its phrasing seems weird to me.

L40. I think the sentence needs to be rewritten, and the word “ecosystem” needs to be placed before “lake water”.

L44. The 2 is missing in 270 km2.

L46-49. I think these two sentences could be merged into one, as they repeat a similar information.

L54. Please add “number” after “species”.

L56. I would remove “which are closely related to the water environment” as it is already stated before.

L57. I think the connection between this sentence and the previous one needs to be made clearer.

L60. Please put an “s” at the end of “cycle” as phytoplankton plays a crucial role in multiple biogeochemical cycles (those of oxygen and carbon, for example).

L61. I think the sentence needs to be reformulated as it is unclear that complex interrelationships refer to relationships between phytoplankton and macrophytes, and not among phytoplankton organisms.

L68. There is a mistake in the species name of the Nymphaea that Chia et al. studied. It is N. lotus, not N. tetragona.

L69-71. Please add some references for this statement.

Material and Methods

L88-89. In this section, I think that the distinction between Lake Caizi, which is the whole lake I believe, and its sub-lakes Caizi, Baitu and Xizi, is not clear enough as I was often confused between terms.

Figure 1. To what refers the term “Caizihu”? To Caizi Lake I presume? In addition, please change the map scale to km (unit of the international system, as requested by the journal’s author recommendations) instead of miles. I would also add to what corresponds each label (BT to Baitu, CZ to Caizi, and XZ to Xizi) in the figure caption.

L112. Please remove the “to collect”.

L113. I believe that chlorophyll a should be spelled with an a in italic. Please modify it here and elsewhere in the manuscript if it is the case.

L116-118. This part needs more information. It does not present sufficient information to be performed again by other. How coverage was assessed by remote sensing?

L119-120. I understand that nutrient and chlorophyll analyses were performed following a standardized method. Maybe it would have been good to have at least a little bit of details about how they were measured.

L123. “Replicates” instead of “repetitions”?

L124. Phytoplankton samples were fixed with Lugol (please add the “l” at the end). What was the final concentration of the lugol solution? In addition, I think that this very long sentence could be reformulated or split.

L132. How was measured cell weight? Please add this information.

L133-134. Maybe you could include the original references for the Shannon-Wiener, Margalef Richness and Pielou Evenness indexes?

L136-142. I have several concerns with this part. First, even if I am not an expert in this field, I believe that in the Margalef Richness Index calculation, N should be the total number of individuals, not density (please see Gamito 2010: Caution is needed when applying Margalef diversity index, Ecol. Ind. 10(2): 550-551. Doi: 10.1016/j.ecolind.2009.07.006). Please ensure that your Margalef index calculations were performed correctly.

Second, I think that ni is missing in the L136, and that fi is missing in the L137. These omissions made my understanding of the formula more complicated than it should be.

In Equation 5, I think that the point should be replaced by an “x” sign, to be consistent with the notation in the other equations.

L154-156. Please add the reference(s) for these empirical formula.

L166-167. How was the normality of the distribution tested? With the Shapiro-Wilk test? In addition, did you test for homoscedasticity (homogeneity of variances)?

Results

L174-175. Please specify that those values are average values form all sampling stations and all sampling dates before or after restoration.

L175. Please remove the double space after “demonstrated”.

L189. “showing that summer > etc..” needs to be reformulated.

L191. “was reduced by 12.92%” Compared to what? Compared to before restoration I presume. Please add it.

L196-197. “only in the spring, the autumn and the winter” could be replaced by “except in summer”.

L197. Please reformulate “had more than Caizi Lake”.

L198. Please put “Which” with a lower case “w”.

L204. “Less” should be replaced by “lower”.

Figure 2. I have multiple remarks concerning figure 2. First, I think that the (l) panel should be put at the same size as the other panels. Second, many information are missing in the figure caption to make sure that the figure is easily understandable. The meaning of upper case A and B (which refer to After restoration and Before restoration I presume) needs to be indicated in the caption. Moreover, it should be indicated that the dashed vertical line separates the before and after restoration periods.

L10. “We investigated the aquatic vegetation in Caizi Lake”. Does this refer to the entire lake (and so all three sub-lakes), or only to the Caizi sub-lake?

L14 and elsewhere in the manuscript. I think “are” should be “were”. On a general note, results should be presented with a past tense, except when it is a general statement. 

L28. Please add a p-value after “significantly” as this word should only be used to describe the result of a statistical test.

Figure 3. Again, please put the map scale in kms instead of miles. I think it is necessary to put in the figure caption to what correspond panels a, b and c. It is unclear for me, I got that a represents the period before restoration (which sample date, an average of all of them?), and b and c represent samples after restorations (but, which sample dates?). Finally, please put the color legend bigger as it is not readable when the manuscript is printed.

L33-52. I think this part needs to be heavily reformulated, as for now it is not well-structured in my opinion. First, past tenses should be used instead of present. Then, I don’t think that the “occupy an absolute advantage” is the right formulation. Finally, I would restructure this entire 3.3.1 part combining both spatial and temporal analyses, and separating in two parts: fist, a description of the cell density and relative abundance before the restoration (including which group is the major one, temporal and spatial variations), then a description after the restoration.

Figure 4. Please add “relative abundance” in the figure caption. In addition, the fact that the dark line represents the cell density needs to be specified as well as the meaning of the error bars (which are standard deviations I assume). Finally, the meaning of the abbreviations in the color legend need to be specified as well in the caption. On a general note, a figure caption need to be precise enough that it is sufficient to understand the figure without having to read the manuscript.

Figure 5. I think there is a misunderstanding in the note explaining the mining of letters. From what I understood, upper case letters indicate significant differences between lakes before restoration, lower case letters indicate significant differences between lakes after restoration, and the comparison between upper and lower case letters for each lake indicate significant differences before and after restoration.

Figure 6. I have similar comments as for Figure 4.

Figure 7. I have similar comments as for the note of Figure 5.

L103. Please change swim plants by phytoplankton.

Figure 8. I have similar comments as for Figure 2.

L141-143. I noticed some mistakes concerning the name of the phytoplankton species. Please correct them. O is missing in Mycrocystis aeruginosa, it is Phormidium tenue and not tenus, the n is missing at the end of Aphanizomenon, and I think that the species name should be flos-aquae, not flosaquae.

Table 2. In the headings, I think “aquatic vegetation” should be replaced by “phytoplankton species”. Moreover, maybe adding a “phytoplankton group” column indicating if the species is a diatom, a cyanobacteria etc. could ease the reading of the table for non-phytoplankton expert readers.

L171. As the p-value is higher than 0.05 (the set significance level) for the correlation between phytoplankton cell density and DO and N/P ratio, it is not significant and should not be listed in the text as nothing can be concluded from it.

L182. The 3.5.2 title needs to be changed.

L183-187. I think that most of this belongs to the Materials and Methods section.

Figure 9. If possible, please show the eigen values lambda 1 and lamda 2 for the axes. I think the environmental factors abbreviations used in the analyses should be spelled out in the figure caption. I noticed a strange black bold line in the panel (a) of the figure, please remove it if it is possible.

Discussion

L239. The effect of aquatic vegetation restoration in Baitu Lake was stated to be the best. Regarding what? Compared to what? In terms of what? I presume it is in terms of vegetation coverage and water quality and compared with the other sub-lakes, but this needs to be made clearer.

L248. “Which” should be replaced by “representing”.

L253. Please reformulate “As we all know”.

L297. To what refers “it”? To aquatic vegetation I presume? Please made it clearer.

L306-307. I think that the sentence refers to all the mechanisms that were mentioned before, so I would change “Therefore” by “In regard to all the above-mentioned mechanisms, …”.

L330-331. This statement is true. However, I think a word is missing, as I don’t understand the meaning of “notably the advantages …” and its connection with the part of the sentence before. It should be reformulated.

L342. I believe that Eichhornia crassipes should be spelled E. crassipes.

L355. It is also the case for cyanobacteria, which have a high surface-to-volume ratio, which is one of the explanation on why cyanobacteria seems to benefit from climate warming in oligotrophic waters. To explain why the relative abundance of green algae increased after the restoration, another hypothesis is missing and should be mentioned: the competition for nutrients among phytoplankton groups. As cyanobacteria abundance decreased after restoration, it could have released the competition pressure on green algae, which allowed them to grow. I think this occurs quite often, at least in marine waters.

Conclusion

I think the conclusion summed up well the major findings of the study. However, I have the feeling that it is too descriptive. I think this study is very “regional”, and I would have liked if results would have been put more in perspective with other lakes, environments (e.g., coastal waters), and other mechanisms of aquatic vegetation (i.e., different from purse seine removal only). Would a different aquatic restoration mechanism have the same effect on the phytoplankton community? Would aquatic vegetation restoration from purse seine removal would have the same effect on other lakes or environments? I understand that all these questions cannot be addressed in the present study, but maybe adding one or two sentences in the conclusion about it would be good.  

References

L455. There is a major problem with Reference n°21. The DOI does not correspond to the right article, which seems very difficult to access for non-Chinese speakers. Please correct the DOI or change the reference.

Author Response

Respose to the Review Comments.

Dear reviewers,

Re: Manuscript ID: diversity-1671136,

Title: Effect of aquatic vegetation restoration after removal of culture purse seine on phytoplankton community structure in Caizi Lakes

We would like to thank you for your careful reading, helpful comments, and constructive suggestions, which has significantly improved the presentation of our manuscript.

We have carefully considered all comments from the reviewers and revised our manuscript accordingly. The manuscript has also been double-checked, and the typos and grammar errors we found have been corrected. We have uploaded a copy of the original manuscript with all the changes highlighted by using the track changes mode in MS Word. Revised portion are marked in red in the paper. The main corrections in the paper and the responds to the reviewer’s comments are as flowing:

In the manuscript entitled “Effect of aquatic vegetation restoration after removal of culture purse seine on phytoplankton community structure in Caizi Lake”, the authors detailed an investigation documenting water quality parameters, aquatic macrophytes distribution and phytoplankton community biomass, composition, diversity, richness and evenness before and after the removal of purse seine in a Chinese lake composed of three sub-lakes, which is directly connected to the middle and lower reaches of the Yangtze River. Globally, this manuscript addresses the effects of the restoration of the lake’s aquatic vegetation from an ecologically very-interesting phytoplankton point of view, which is often under-studied compared to more basic analyses of water quality and macrophytes coverage studies. Using 12 samplings spanning over almost five years and at 20 different stations within the lake, the authors reported important changes in phytoplankton biomass and community composition which they related to changes in water quality and macrophytes coverage induced by the purse seine removal. The study was well-designed and enabled the detection of the consequences of the removal of purse seine on phytoplankton. However, I found that both the Materials and Methods and the Results sections need heavy revisions as many parts are unclear and/or incomplete. In addition, most figure captions are not precise enough and do not present sufficient information to fully understand the figures they are referring to. The Introduction and Discussion sections seem quite well-structured, although some paragraphs could be shortened and better-structured. Nonetheless, the Discussion section relates well the reported results with other studies. The Conclusion section sums up well the major findings of the study, however I think it is too descriptive, and more contextualization and generalization should be done. Finally, many small errors greatly reduce the overall quality of the manuscript. In this regard, I strongly recommend to use a professional language revision service. I have numerous specific comments that are listed below. 

Response: Thank you for your valuable and thoughtful comments. Based on your suggestions, we have carefully revised the manuscript, and the revisions mainly include: (1) Revised the Materials and Methods section and the Results section. (2) Added the missing information in the captions of the figure or table. (3) The summary section was rewritten. (4) Revised the typos and grammar errors in the text. (5) Rewrote sentences that did not make sense or were poorly presented.

Abstract

L15, 16, 17, 18. I think it is necessary to specify that the values presented represent average of all the stations and all the sampling dates before and after the restoration.

Response: We deeply appreciate your suggestion. According to the reviewer’s comment, we have added a more detailed interpretation regarding the values provided, as follows: The results showed that the average dissolved oxygen (from 7.43±0.25 mg/L to 9.12±0.49 mg/L) and Secchi depth (from 28.40±6.20 cm to 47.61±14.62 cm) in the water column of Caizi Lakes increased after the restoration of aquatic vegetation, while the average concentrations of total nitrogen (from 2.00±0.16 mg/L to 1.34±0.18 mg/L) and total phosphorus (from 0.15±0.02 mg/L to 0.06±0.01 mg/L) decreased. (L13-18).

Introduction

L28. Please replace “are” by “is”.

Response: We appreciate you pointing out this issue. In the revised manuscript, “are” (L29) has been replaced by “is” (L29).

L31-34. I think that this sentence could be spited to 2 sentences, that would make it easier to read.

Response: Thank you for the above suggestion. We have modified the sentence according to the comment, as follows: Since 2010, purse seine culture in lakes connected to the Yangtze River has increased dramatically, 80% of the water surface has been covered by purse seine. The increased density of fish such as herbivorous fish threatened the survival of aquatic vegetation, especially the severe degradation of submerged vegetation. (L33-37)

L35. I would change “deposit” for “settle”.

Response: Thank you for pointing out this problem in manuscript. In the revised manuscript, we replaced “deposit” with “settled”. (L39)

L37. This sentence should be reformulated as its phrasing seems weird to me.

Response: Thank you for your rigorous advice. This sentence was rephrased according to the comment, as follows: In addition, pollution from aquaculture will be superimposed within the seine area due to the barrier effect of the seine, so long-term and high-density of seine farming will lead to eutrophication of the lake [4]. (L40-43)

L40. I think the sentence needs to be rewritten, and the word “ecosystem” needs to be placed before “lake water”.

Response: Thank you for your rigorous consideration. Followed your suggestion, we modify this sentence to “In order to restore the structure and function of the aquatic ecosystem of Yangtze River-connected lake”. (L43-44)

L44. The 2 is missing in 270 km2.

Response: Thank you so much for your careful check. In the revised manuscript, we have revised “270 km” to “270 km2”. (L47)

L46-49. I think these two sentences could be merged into one, as they repeat a similar information.

Response: We gratefully appreciate for your valuable suggestion. Based on your suggestion, we have combined the above two sentences into one sentence as “Numerous studies have shown that the removal of aquaculture seines is beneficial to the restoration of aquatic vegetation diversity.” (L49-50)

L54. Please add “number” after “species”.

Response: Thank you for underlining this deficiency. As per your suggestion, we have added "number" after "species" in the revised manuscript. (L55)

L56. I would remove “which are closely related to the water environment” as it is already stated before.

Response: Thank you for the suggestion. We have removed the phrase “which are closely related to the water environment”. (L58)

L57. I think the connection between this sentence and the previous one needs to be made clearer.

Response: We gratefully appreciate for your valuable comment. Following your suggestion, we modified the sentence (L56-59) to “Aquatic vegetation is of great significance to the stability of aquatic ecosystem. It is not only a key link in the food chain of aquatic ecosystem, but also increases the water transparency, stabilizes sediment and reduces water velocity [7].”

L60. Please put an “s” at the end of “cycle” as phytoplankton plays a crucial role in multiple biogeochemical cycles (those of oxygen and carbon, for example).

Response: Thank you for pointing out this problem in manuscript. We replaced “cycle” with “cycles” in the revised manuscript. (L61)

L61. I think the sentence needs to be reformulated as it is unclear that complex interrelationships refer to relationships between phytoplankton and macrophytes, and not among phytoplankton organisms.

Response: Thank for your comment. We agree with the comment and re-wrote the sentence in the revised manuscript as the following: There are complex interrelationships between aquatic vegetation and phytoplankton, such as competition [9], allelopathy [10]. (L61-63)

L68. There is a mistake in the species name of the Nymphaea that Chia et al. studied. It is N. lotus, not N. tetragona.

Response: Thank you for your careful review. We changed “Nymphaea tetragona " into “Nymphaea lotus” in the revised manuscript. (L69)

L69-71. Please add some references for this statement.

Response: Thank you for the suggestion. This statement was obtained from a field survey, so we did not add a reference. Based on your concern, we considered that it may be due to the lack of detail in the description of the time of the survey, so we revised this statement to: Our investigation revealed that floating plants, mainly T. bispinosa and Eichhornia crassipes, recovered rapidly after the removal of the aquaculture seine in Caizi Lakes, especially in Lake Baitu, where 40% of the water surface was covered by aquatic vegetation. (L71-74)

Material and Methods

L88-89. In this section, I think that the distinction between Lake Caizi, which is the whole lake I believe, and its sub-lakes Caizi, Baitu and Xizi, is not clear enough as I was often confused between terms.

Response: We feel sorry for the inconvenience brought to you. In order to clearly distinguish between the entire lake and the sub-lakes of Lake Caizi, we redefine the whole lake as “Caizi Lakes”, and the sub-lakes of Caizi Lakes as “Lake Baitu, Lake Caizi and Lake Xizi”. (L91-93)

Figure 1. To what refers the term “Caizihu”? To Caizi Lake I presume? In addition, please change the map scale to km (unit of the international system, as requested by the journal’s author recommendations) instead of miles. I would also add to what corresponds each label (BT to Baitu, CZ to Caizi, and XZ to Xizi) in the figure caption.

Response: Thank you for the above suggestions. According to your suggestion, we have made the following modifications in the revised manuscript: (1) We are very sorry for our negligence of term “Caizihu”, and we have corrected “Caizihu” to “Caizi Lakes” (L115). (2) We changed the unit of map scale from miles to kilometers of international system (L114). (3) We have added the meaning of the letters of the diagram in the title, as follows: “BT to Baitu, CZ to Caizi, and XZ to Xizi”. (L115-116)

L112. Please remove the “to collect”.

Response: Thank you so much for your careful check. We have removed “to collect” from the revised manuscript. (L122)

L113. I believe that chlorophyll a should be spelled with an a in italic. Please modify it here and elsewhere in the manuscript if it is the case.

Response: Thank you for pointing out this problem in manuscript. As per your suggestion, we have italicized the “a” in “chlorophyll a” throughout the manuscript.  (L123)

L116-118. This part needs more information. It does not present sufficient information to be performed again by other. How coverage was assessed by remote sensing?

Response: Thank for your comment. We added the above required information in the revised manuscript as follows: We surveyed the species and distribution of aquatic plants in Caizi Lakes by visual boating. In addition, we also collected remote sensing images from Sentinel-2 MSI of Caizi Lakes in July 2017, August 2019 and July 2021 (http://slj.anqing.gov.cn). The image has less cloud cover, high quality and 10 m ground resolution. We processed remote sensing image data by The Environment for Visualizing Images (version 5.5), used the decision tree classification to distinguish vegetated areas from non-vegetated areas, and calculated the coverage of vegetated areas (Cov) [19]. (L126-133)

L119-120. I understand that nutrient and chlorophyll analyses were performed following a standardized method. Maybe it would have been good to have at least a little bit of details about how they were measured.

Response: Thank you for your suggestion. We added the determination methods of nutrients and chlorophyll a concentration as follows: Total nitrogen (TN) was determined by alkaline potassium persulfate digestion UV spectrophotometric method, total phosphorus (TP) by ammonium molybdate spectrophotometric method, ammonia nitrogen (NH4+-N) by nessler’s reagent spectrophotometry and nitrate nitrogen (NO3--N) by Ultraviolet spectrophotometry, and Chl. a by acetone extraction spectrophotometry under an ultraviolet spectrophotometer (UV 2450). (L134-140)

L123. “Replicates” instead of “repetitions”?

Response: Thank you so much for your careful check. We changed “repetitions” to “replicates” in the revised manuscript. (L140)

L124. Phytoplankton samples were fixed with Lugol (please add the “l” at the end). What was the final concentration of the lugol solution? In addition, I think that this very long sentence could be reformulated or split.

Response: Thank you for the above suggestions. Modifications to this question are as follows: Quantitative phytoplankton samples were fixed with Lugol iodine solution at a fixation dose of 1% of the sample volume. We removed the supernatant slowly by siphoning after they had precipitated for at least 48 h until concentrated to 30 mL. After mixing, we took 0.1 ml samples and placed them in a plankton counting chamber [6], and randomly selected 100 visual fields under × 400 times optical microscope (BX53, Olympus) to classify and count species [21]. (L141-146)

L132. How was measured cell weight? Please add this information.

Response: Thank you for the suggestion. We added the information required above as follows: We counted the number of phytoplankton cells and calculated their cell density and biomass. Biomass was obtained by multiplying the number of cells and the weight of cells. Since the specific gravity of phytoplankton is close to 1, we directly converted the volume to weight, and the cell volume was determined by the average cell dimensions of each species. (L148-152)

L133-134. Maybe you could include the original references for the Shannon-Wiener, Margalef Richness and Pielou Evenness indexes?

Response: Thank for your suggestion. According to your suggestion, we have added corresponding references to the Shannon-Wiener, Margalef and Pielou indexes. (L153-154)

L136-142. I have several concerns with this part. First, even if I am not an expert in this field, I believe that in the Margalef Richness Index calculation, N should be the total number of individuals, not density (please see Gamito 2010: Caution is needed when applying Margalef diversity index, Ecol. Ind. 10(2): 550-551. Doi: 10.1016/j.ecolind.2009.07.006). Please ensure that your Margalef index calculations were performed correctly.

Second, I think that ni is missing in the L136, and that fi is missing in the L137. These omissions made my understanding of the formula more complicated than it should be.

In Equation 5, I think that the point should be replaced by an “x” sign, to be consistent with the notation in the other equations.

Response: Thank you for the above suggestions. We apologize for the inconvenience caused by our oversight in reviewing your manuscript.

(1) This sentence was rephrased according to the comment. Where ni is the number of individuals of species i, N is the total number of individuals of all species, fi is the frequency of individuals in the i species, S is the total number of phytoplankton species, and ni/N represents the relative proportion of species i. (L159-162)

(2) Based on your suggestion, we replaced this point with the “×” sign in equation 5. (L170)

L154-156. Please add the reference(s) for these empirical formula.

Response: Thank you for the suggestion. We added references to these formulas in the revised manuscript. (L169)

L166-167. How was the normality of the distribution tested? With the Shapiro-Wilk test? In addition, did you test for homoscedasticity (homogeneity of variances)?

Response: We totally understand your concerns. As you predicted, we verified whether the data satisfied the normal distribution by using the Shapiro-Wilk test. In addition, we also tested the homogeneity of variance of data before one-way ANOVA. The above related contents have been supplemented in the revised manuscript. (L195-196)

Results

L174-175. Please specify that those values are average values form all sampling stations and all sampling dates before or after restoration.

Response: Thank you for pointing out this problem in manuscript. According to your suggestion, we added the mean values of physicochemical indicators for all sampling stations and all sampling dates before or after aquatic vegetation restoration. (L205-L206)

L175. Please remove the double space after “demonstrated”.

Response: Thank you so much for your careful check. We removed the extra space after “demonstrated”. (L206)

L189. “showing that summer > etc..” needs to be reformulated.

Response: Thank you for your rigorous comment. We modified the above sentence to “In addition, Chl. a was higher in summer and autumn than in spring and winter.” (L219-220)

L191. “was reduced by 12.92%” Compared to what? Compared to before restoration I presume. Please add it.

Response: Thank you for underlining this deficiency. This sentence was rephrased according to the comment: The TLI of aquatic vegetation before and after restoration was 61.59±0.67 and 53.58±0.71, respectively, which indicated that Caizi Lakes changed from moderate eutrophication to mild eutrophication. (L220-223)

L196-197. “only in the spring, the autumn and the winter” could be replaced by “except in summer”.

Response: Thank for your comment. As you suggested, the phrase “except in summer” to describe “only in the spring, the autumn and the winter” would be more concise. However, since your suggestion brought this problem to our attention, we have deleted this phrase by combining the following and the discussion and carefully considering it, so as to make the result description more concise. (L227-229)

L197. Please reformulate “had more than Caizi Lake”.

Response: We are very grateful to you for pointing out this problem. The sentence was rewritten to read: However, the SD of Lake Baitu in summer was higher than that of Lake Caizi and Lake Xizi, by 9.23% and 4.39%, respectively, corresponding to values of 75.02±5.25 cm, 68.09±4.32 cm and 71.73±4.26 cm, respectively (Fig. 2d).  (L227-229)

L198. Please put “Which” with a lower case “w”.

Response: Thank you so much for your careful check. According to the previous suggestion, we modified the sentence without “which”. (L227-229)

L204. “Less” should be replaced by “lower”.

Response: Thank you for underlining this deficiency. According to your suggestion, we changed “less” to “lower”. (L236)

Figure 2. I have multiple remarks concerning figure 2. First, I think that the (l) panel should be put at the same size as the other panels. Second, many information are missing in the figure caption to make sure that the figure is easily understandable. The meaning of upper case A and B (which refer to After restoration and Before restoration I presume) needs to be indicated in the caption. Moreover, it should be indicated that the dashed vertical line separates the before and after restoration periods.

Response: Thank you for the above suggestions. The changes are as follows:

(1) We modified the size of the (l) panel in order to make each panel the same size. (L241-242)

(2) We added relevant information missing from the graphic captions, including: the index of the graph, the meaning of the capital A and B letters and the meaning of the dashed vertical line, as follows: Abbreviations was used in the diagram: (a) WT: water temperature, (b) WD: water depth, (c) pH, (d) SD: Secchi depth, (e) DO: dissolved oxygen, (f)Turb: turbidity, (g) TP: total phosphorus, (h) TN: total nitrogen, (i) AN: ammonia nitrogen, (j) XN: nitrate nitrogen, (k) Chl. a: chlorophyll a, and (l) TLI: comprehensive nutrient status index. The capital letters B and A represent before and after the restoration of aquatic vegetation, respectively. Vertical lines separated the before (October 2016 to July 2017) and after (August 2019 to July 2021) restoration periods. (L243-249)

L10. “We investigated the aquatic vegetation in Caizi Lake”. Does this refer to the entire lake (and so all three sub-lakes), or only to the Caizi sub-lake?

Response: Thank for your comment. The word “Caizi Lake” in this sentence refers to the entire lake. In order to better distinguish between the entire lake and its sub-lakes, we have defined the entire lake as the Caizi Lakes and changed it in the full text. (L251)

L14 and elsewhere in the manuscript. I think “are” should be “were”. On a general note, results should be presented with a past tense, except when it is a general statement. 

Response: Thank you for pointing out this problem in manuscript. According to your comment, we changed the sentence “are” to “were” and corrected other similar errors in the manuscript. (L257)

L28. Please add a p-value after “significantly” as this word should only be used to describe the result of a statistical test.

Response: Thank you for the suggestion. In the manuscript, we added the significant P value of plant coverage before and after aquatic vegetation restoration. (L273)

Figure 3. Again, please put the map scale in kms instead of miles. I think it is necessary to put in the figure caption to what correspond panels a, b and c. It is unclear for me, I got that a represents the period before restoration (which sample date, an average of all of them?), and b and c represent samples after restorations (but, which sample dates?). Finally, please put the color legend bigger as it is not readable when the manuscript is printed.

Response: Based on your suggestions, we have made the following changes. (1) We modified the map scale units to kilometers (L274-275). (2) We added the sampling dates corresponding to a, b and c to the caption of Fig. 3, as follows: Panel(a) presented the aquatic vegetation in July 2017 before restoration, and panels(b) and (c)presented the aquatic vegetation in August 2019 and July 2021 after restoration, respectively(L276-278). (3) We have enlarged the legend appropriately to make the picture clearer (L274-L275).

L33-52. I think this part needs to be heavily reformulated, as for now it is not well-structured in my opinion. First, past tenses should be used instead of present. Then, I don’t think that the “occupy an absolute advantage” is the right formulation. Finally, I would restructure this entire 3.3.1 part combining both spatial and temporal analyses, and separating in two parts: fist, a description of the cell density and relative abundance before the restoration (including which group is the major one, temporal and spatial variations), then a description after the restoration.

Response: We are very grateful to you for taking your valuable time to put forward constructive remarks. According to your above suggestions, we have made the following modifications: (1) We corrected the description of the present tenses to the past tenses. (2) The phrase “occupy an absolute advantage” has been revised to “occupied the main advantage”. (3) We think your proposal will make the article more complete, but since our study mainly explores spatial and temporal variation, in addition, Figures 4 and 5 were also based on spatial and temporal variation, we apologize for not dividing the description of Section 3.3.1 directly into two parts before and after the restoration of aquatic vegetation. However, the modification were also based on the change of phytoplankton cell density before and after the restoration of aquatic vegetation. (L282-299)

Figure 4. Please add “relative abundance” in the figure caption. In addition, the fact that the dark line represents the cell density needs to be specified as well as the meaning of the error bars (which are standard deviations I assume). Finally, the meaning of the abbreviations in the color legend need to be specified as well in the caption. On a general note, a figure caption need to be precise enough that it is sufficient to understand the figure without having to read the manuscript.

Response: We deeply appreciate your suggestion on the graphic title. According to your suggestion, we modified the title of Figure 4 as follows: (1) The left axis or bar graph indicated the relative abundance of phytoplankton cell density, and the right axis or dark line indicated the average cell density (average and standard deviation) of all points at each sampling date in Caizi Lakes. (2) Abbreviations was used in the diagram: Pyr: Pyrrophyta, Xan: Xanthophyta, Chr: Chrysophyta, Eug: Euglenophyta, Cry: Cryptophyta, Bac: Bacillariophyta, Chl: Chlorophyta, Cya: Cyanobacteria. (3) Capital letters B and A indicate before and after aquatic plant restoration, respectively. (L306-312)

Figure 5. I think there is a misunderstanding in the note explaining the mining of letters. From what I understood, upper case letters indicate significant differences between lakes before restoration, lower case letters indicate significant differences between lakes after restoration, and the comparison between upper and lower case letters for each lake indicate significant differences before and after restoration.

Response: We feel sorry for the trouble brought to you. As you understand, upper case letters indicate significant differences between lakes before restoration, lower case letters indicate significant differences between lakes after restoration. The comparison between upper and lower case letters for each lake indicated significant differences before and after restoration. The same letter indicated a non-significant difference(P>0.05), different letters indicated a significant difference (P < 0.05). We have added the above explanation to the figure title. (L314-319)

Figure 6. I have similar comments as for Figure 4.

Response: Thank for your comment. We have added to the figure captions the meanings represented by the left and right vertical axes, which differ from the caption of Figure 4 in that the vertical axes represent information related to phytoplankton biomass. We have added the specific phylum represented by the abbreviations. In addition, we have also added the meaning of the capital letters A and B. (L338-344)

Figure 7. I have similar comments as for the note of Figure 5.

Response: Thank for your comment. Based on your suggestion, similar to the changes made to the title of Figure 5, we have added the meanings represented by the various letters in the figure title. (L357-362)

L103. Please change swim plants by phytoplankton.

Response: We are sorry for our mistake in writing. We have corrected “swim plants” to “phytoplankton” in the revised manuscript. (L363)

Figure 8. I have similar comments as for Figure 2.

Response: Thank you for the above suggestions. Detailed explanations were supplemented, as follows. (1) We adjusted the size of the panels to keep them consistent in size. (2) We supplemented the indexes of the images, the letters A and B and the significance of the vertical dashed lines as follows, panels a, b and c represented the Shannon-Wiener diversity index of phytoplankton, the Margalef index and the Pielou index, respectively. Capital letters B and A represent the before and after restoration of aquatic vegetation, respectively. In addition, vertical lines separated the period before (October 2016 to July 2017) and after (August 2019 to July 2021) recovery. (L398-402)

L141-143. I noticed some mistakes concerning the name of the phytoplankton species. Please correct them. O is missing in Mycrocystis aeruginosa, it is Phormidium tenue and not tenus, the n is missing at the end of Aphanizomenon, and I think that the species name should be flos-aquae, not flosaquae.

Response: Thank you so much for your careful check. Based on your comments, we have corrected the spelling of Mycrocystis aeruginosa, Phormidium tenue and Aphanizomenon flos-aquae. (L406-L409)

Table 2. In the headings, I think “aquatic vegetation” should be replaced by “phytoplankton species”. Moreover, maybe adding a “phytoplankton group” column indicating if the species is a diatom, a cyanobacteria etc. could ease the reading of the table for non-phytoplankton expert readers.

Response: We gratefully appreciate for your valuable suggestion. In the headings, we changed “aquatic plants” to “phytoplankton species”. In addition, according to your proposal, we added a column of “phytoplankton groups” to indicate the phylum to which the species belongs. (L423-424)

L171. As the p-value is higher than 0.05 (the set significance level) for the correlation between phytoplankton cell density and DO and N/P ratio, it is not significant and should not be listed in the text as nothing can be concluded from it.

Response: Thank for your comment. Based on your comments, we have removed the above sentence. (L428)

L182. The 3.5.2 title needs to be changed.

Response: Thank you for pointing out this problem in manuscript. According to your proposal, we thought that “RDA” and “Redundancy Analysis” were reused. However, after careful consideration, we have removed the titles “3.5.1 Pearson Correlation Analysis” and “3.5.2 RDA Redundancy Analysis”. This is because both chapters described the relationship between phytoplankton communities and environmental factors, and the two removed headings simply expressed two different methods of analysis, which were mentioned in the Materials and Methods section. Furthermore, heading 3.5 already indicated to the reader the subject of this chapter. (L441)

L183-187. I think that most of this belongs to the Materials and Methods section.

Response: We are grateful for the suggestion. Following your suggestion, we have added this section to the Materials and Methods section after organizing it. (L190-L193)

Figure 9. If possible, please show the eigen values lambda 1 and lamda 2 for the axes. I think the environmental factors abbreviations used in the analyses should be spelled out in the figure caption. I noticed a strange black bold line in the panel (a) of the figure, please remove it if it is possible.

Response: Thank for your comments. The modifications are as follows. We corrected the following. (1) We added eigen values lambda 1 and lamda 2 for the axes, and described its explanation rate in the results. (2) We added environmental factors abbreviations in the figure caption. (3) We replaced the previous unsuitable panel a. (L462-470)

Discussion

L239. The effect of aquatic vegetation restoration in Baitu Lake was stated to be the best. Regarding what? Compared to what? In terms of what? I presume it is in terms of vegetation coverage and water quality and compared with the other sub-lakes, but this needs to be made clearer.

Response: We gratefully thanks for you spent the precious time to provide constructive remarks. Compared with the restoration coverage of aquatic vegetation in Lake Caizi and Lake Xizi, the restored cover of aquatic vegetation Baitu in Lake was the most, so we judged that the restoration effect of aquatic vegetation in Baitu Lake was the best. (L) This sentence was rephrased according to the comment, We have re-written this part according to the your suggestion. In addition, we added to the results the specific area of aquatic plants restored in the sub-lake in question. (L488-L492)

L248. “Which” should be replaced by “representing”.

Response: Thank you so much for your careful check. Based on your suggestion, “Which” was replaced by “representing” in the revised manuscript. (L501)

L253. Please reformulate “As we all know”.

Response: We are grateful for the suggestion. We modified “As we all know” to “It is well known that”. (L506)

L297. To what refers “it”? To aquatic vegetation I presume? Please made it clearer.

Response: Thank for your comments. We have already explained in the manuscript, that “it” in this context refers to “aquatic vegetation”. (L549)

L306-307. I think that the sentence refers to all the mechanisms that were mentioned before, so I would change “Therefore” by “In regard to all the above-mentioned mechanisms, …”.

Response: Thank you for pointing out this problem. We have revised this sentence according to your opinion. (L559)

L330-331. This statement is true. However, I think a word is missing, as I don’t understand the meaning of “notably the advantages …” and its connection with the part of the sentence before. It should be reformulated.

Response: We gratefully appreciate for your valuable suggestion. This sentence has been rewritten as: This was in general agreement with the results of others, where aquatic vegetation restoration significantly reduced the relative abundance of Cyanophyta and rose the relative abundance of Chlorophyta, Bacillariophyta and Cryptophyta in the Huayang lakes. (L582-583)

L342. I believe that Eichhornia crassipes should be spelled E. crassipes.

Response: Thank you so much for your careful check. We have modified“Eichhornia crassipes”to “E. crassipes”. (L595-596)

L355. It is also the case for cyanobacteria, which have a high surface-to-volume ratio, which is one of the explanation on why cyanobacteria seems to benefit from climate warming in oligotrophic waters. To explain why the relative abundance of green algae increased after the restoration, another hypothesis is missing and should be mentioned: the competition for nutrients among phytoplankton groups. As cyanobacteria abundance decreased after restoration, it could have released the competition pressure on green algae, which allowed them to grow. I think this occurs quite often, at least in marine waters.

Response: Thank you for your rigorous advice. According to your opinion, we have added the following contents: Moreover, after the restoration of aquatic vegetation, the number of Cyanophyta decreased, which may release the competitive pressure on Chlorophyta, thus grew it. (L611-L613)

Conclusion

I think the conclusion summed up well the major findings of the study. However, I have the feeling that it is too descriptive. I think this study is very “regional”, and I would have liked if results would have been put more in perspective with other lakes, environments (e.g., coastal waters), and other mechanisms of aquatic vegetation (i.e., different from purse seine removal only). Would a different aquatic restoration mechanism have the same effect on the phytoplankton community? Would aquatic vegetation restoration from purse seine removal would have the same effect on other lakes or environments? I understand that all these questions cannot be addressed in the present study, but maybe adding one or two sentences in the conclusion about it would be good. 

Response: Thanks very much for taking your time to review this manuscript. Based on your suggestions, we have revised our conclusions as follows: In this study, we explored the response of phytoplankton to aquatic vegetation recovery after the removal of purse seine. The results showed that aquatic vegetation recovered in Caizi Lakes after the removal of farming seine nets, which improved lake water quality and drove the evolution of phytoplankton community structure. Our results were consistent with the performance of some other shallow lakes such as Huayang Lakes. According to the theory of alternative stable states in shallow lakes, this state of transformation from turbid water to clear water in Caizi Lakes after the removal of the seine is predicted. When the anthropogenic disturbance was removed, the lake naturally recovers and tended to a stable structure. Therefore, it is recommended that the lake seine be completely removed to reduce the anthropogenic disturbance in the lake. This study was conducted with a view to providing a basic theoretical basis for evaluating the restoration effect of aquatic vegetation from the perspective of phytoplankton, which was indicative of lake protection and ecological restoration. However, whether the role of aquatic vegetation on aquatic environmental restoration is consistent in different environments such as the ocean or reservoirs is still unknown and needs further analysis. In addition, when we explored the effect of seine removal on phytoplankton community structure, only the biological factor aquatic vegetation was selected, which can be further explored in the follow-up work in combination with bird, fish and zooplankton changes. (L671-L689)

References

L455. There is a major problem with Reference n°21. The DOI does not correspond to the right article, which seems very difficult to access for non-Chinese speakers. Please correct the DOI or change the reference.

Response: Thank you so much for your careful check. We modified the DOI of the above reference. (L782-783)

Thank you again!

Author Response

Respose to the Review Comments.

Dear reviewer,

Re: Manuscript ID: diversity-1671136,

Title: Effect of aquatic vegetation restoration after removal of culture purse seine on phytoplankton community structure in Caizi Lakes

Authors: Wenqian Zhao, Zhenzhong Liu, Wenli Guo and Zhongze Zhou

We would like to thank you for your careful reading, helpful comments, and constructive suggestions, which has significantly improved the presentation of our manuscript.

We have carefully considered all comments from the reviewers and revised our manuscript accordingly. The manuscript has also been double-checked, and the typos and grammar errors we found have been corrected. We have uploaded a copy of the original manuscript with all the changes highlighted by using the track changes mode in MS Word. Revised portion are marked in red in the paper. The main corrections in the paper and the responds to the reviewer’s comments are as flowing:

The manuscript addresses an interesting topic concerning the response of phytoplankton community structure to aquatic vegetation restoration after purse seine removal in Caizi Lake, China.

The results of these studies were very predictable, I mean that after the removal of purse seine and the development of aquatic vegetation, the abundance and biomass of phytoplankton would decrease and phytoplankton taxonomic structure would change. This is how Scheffer's alternative stable states work, this phenomenon is very widely described in the literature. I am surprised that the phenomenon of alternative states has not been addressed at all in this work. Nevertheless, I think that the description of the results and the situation before and after restoration in the lake has scientific value.

The authors illustrated changes in the phytoplankton structure in relation to the restoration of aquatic vegetation using commonly used diversity indices such as Shannon-Wiener, diversity, Margalef richness and Pielou evenness index.

I noticed a mixing of tenses throughout the text. The present tense (and also Present Perfect in some cases) was often used to present facts from the closed past.

Other comments I presented below.

Response: Thank you for your valuable and thoughtful comments. Based on your suggestions, we have carefully revised the manuscript, and the revisions mainly include: (1) Revised the Materials and Methods section and the Results section. (2) Added the missing information in the captions of the figure or table. (3) The summary section was rewritten. (4) Revised the typos and grammar errors in the text. (5) Rewrote sentences that did not make sense or were poorly presented. (6) Added discussion about the phenomenon of alternative states.

Introduction

line 67: Scenedesmus should be written in italics.

Response: Thank for your comment. We have italized “Scenedesmus”. (L69)

Materials and Methods

line 124: “…with Lugo iodine solution… - should be Lugol

Response: Thank you for pointing out this problem in manuscript. We replaced “Lugo” with “Lugol” in the revised manuscript. (L141)

lines 123-126: “We fixed phytoplankton samples with Lugo iodine solution, precipitated for at least 48 hours, slowly removed supernatant by siphon until concentrated to 30mL, then mixed evenly, took 0.1 mL samples and placed them in plankton counting chamber.” - If the studied lake is moderate eutrophic, it’s hard to imagine how to count phytoplankton “objects” after concentration the sample from 1L to 30 mL. Could the Authors explain how they did it? 

Response: We removed the supernatant slowly by siphoning after they had precipitated for at least 48 h until concentrated to 30 mL. After mixing, we took 0.1 ml samples and placed them in a plankton counting chamber, and randomly selected 100 visual fields under × 400 times optical microscope to classify and count species. We counted two 0.1mL pieces of the concentrated sample and then averaged them.

Results

My concern is how it is possible that the lake described is moderately eutrophic (before purse seine removal), according to the authors, can have such low values for chlorophyll and other parameters. These data seem quite puzzling. In addition, the differences before and after removal of purse seine are not significantly large. When comparing the results (except for the biogens values) from the different seasons: summer, autumn, spring, they are very similar season to season.

Besides, in the description of the results, the authors use mainly percentage values (the

table contains quantitative values). In my opinion, this is not correct, as the percentage

values do not fully reflect the situation in the lake and may give a misleading idea of the whole situation.

Response: Thank for your comments. We totally understand your concern.

(1) We determined the nutrient status of the lake based on the integrated trophic state index method (TLI), and calculated TLI values from TN, TP, SD, Chl. a and COD. When TLI is between 50-60 lakes are Mild eutrophication and between 60-70 are Moderate eutrophication. in our study, TLI was 61.59 before the aquatic plants were restored, so Caizi lakes was judged to be moderate eutrophication. According to the water quality evaluation criteria, the lake is considered eutrophic when the total phosphorus is higher than 0.11 mg/L, or the total nitrogen concentration is higher than 1.2 mg/L, or the transparency is 0.55m. In our study, most of the data before the restoration of aquatic plants were satisfied with the eutrophic state of the water body. However, we have noticed before that the Chl. a concentration was low, but the sampling method and experimental method were correct, we speculated that this can be caused by the predation effect of zooplankton or by the interaction between phytoplankton and microorganisms. We cannot explain this phenomenon well yet, and we also found low Shengjin Lake, China. Therefore, we need to further study the relationship between phytoplankton and Chl. a.

(2) We derived from one-way ANOVA that pH, SD, DO, TN, TP and AN were highly significant (P < 0.01) and Turb and Chl. a were significantly different (P < 0.05) before and after aquatic vegetation restoration.

(3) Based on your suggestion, we have changed some of the percentage values to quantitative values. However, to make it easier to see the change before and after recovery, we have still used percentage values in some cases.

line 8: Figure 2. Spatial variation of different physical and chemical indexes. These graph don’t present indexes. Please add appropriate title.

Response: Thank you for the above suggestions. The changes are as follows: We added relevant information missing from the graphic captions, as follows: Abbreviations was used in the diagram: (a) WT: water temperature, (b) WD: water depth, (c) pH, (d) SD: Secchi depth, (e) DO: dissolved oxygen, (f)Turb: turbidity, (g) TP: total phosphorus, (h) TN: total nitrogen, (i) AN: ammonia nitrogen, (j) XN: nitrate nitrogen, (k) Chl. a: chlorophyll a, and (l) TLI: comprehensive nutrient status index. The capital letters B and A represent before and after the restoration of aquatic vegetation, respectively. (L243-249)

line 31: The title of Figure 3 is unclear. Should be written which graph presents which season etc.

Response: Response: Based on your suggestions, we have made the following changes. We added the sampling dates corresponding to a, b and c to the caption of Fig. 3, as follows: Panel(a) presented the aquatic vegetation in July 2017 before restoration, and panels(b) and (c)presented the aquatic vegetation in August 2019 and July 2021 after restoration, respectively. (L276-278)

lines 41-44: “…density of chlorophyta, diatom and euglena increased in…” - Please use appropriate names of groups of phytoplankton. Please decide what you’re talking about: genera, the whole groups, in English or in Latin. Use italics where needed.

Response: We revised “chlorophyta, diatom and euglena” in the above sentences to “Chlorophyta, Bacillariophyta and Euglenophyta”, and corrected them throughout the text. (L290)

line 65: What do the letters A, B, a, b in the figures 5 and 7 mean?

Response: We feel sorry for the trouble brought to you. As you understand, upper case letters indicate significant differences between lakes before restoration, lower case letters indicate significant differences between lakes after restoration. The comparison between upper and lower case letters for each lake indicated significant differences before and after restoration. The same letter indicated a non-significant difference(P>0.05), different letters indicated a significant difference (P < 0.05). We have added the above explanation to the figure title. (L314-319)

lines 71-73: “…abundance of cyanobacteria shrank from 46.70% to 12.12% (P < 71 0.01). After the restoration of the aquatic vegetation, the relative abundance of diatoms 72 and green algae increased by 19.73% (P < 0.01) and 10.00% (P < 0.01), respectively, and 73 that of euglena and cryptoalgae climbed by 4.27% and 0.85%, respectively…”  Where do these percentages come from?

Are these any average numbers for seasons or something different?

Response: Thank for your comments. The values in the above sentences were calculated based on the average of the seasons. For example, to obtain the relative abundance data of cyanobacterial biomass, we counted the average biomass of cyanobacteria in all sample sites before the restoration of aquatic vegetation in four seasons respectively, and then calculated the average of seasons, i.e., averaged into four parts to obtain the average biomass of cyanobacteria before restoration, and its relative abundance was expressed as the ratio of the average biomass of cyanobacteria to the average biomass of all phytoplankton. After the removal of the seine, the mean biomass was also calculated based on the seasons, but due to the epidemic, the seasons sampled were unevenly distributed, including three summers, so we considered the summer phytoplankton biomass as the mean of the three summer biomasses.

lines 87-88: “However, the annual average biomass of Caizi Lake is always at the highest peak before and after the…” What do you mean saying “the annual average biomass”? What does“annual” mean? Before the aquatic plants lush development, there were one sampling in autumn 2016 and three samplings in 2017, so I don’t know what “annual” means. The same for the rest of the survey - two samplings in 2019 and two in 2021. Do we have the right to talk about annual values based on 2 samples made in a given year? In my opinion, we don't.

Response: Thank you for pointing out this problem in manuscript. Based on your suggestion, we found that the term "annual average biomass" is wrong. Therefore, we changed the term "annual average biomass" to "average biomass", which refers to the average biomass before or after the removal of the phytoplankton seine, and this value is a seasonal average. (L321)

line 103: “of swim plants”? I’m confused.

Response: We are sorry for our mistake in writing. We have corrected “swim plants” to “phytoplankton” in the revised manuscript. (L363)

lines 141-146: What was the method of dominant species designation? Should be described in the section of Materials & Methods. In this point I’d like to mention that some of the names of species are wrongly written and some are out of date now. I think that the journal about diversity of phytoplankton needs to give current names of species. This note applies to the whole manuscript. Please check algaebase.org website.

Response: Thank for your comments. In response to your suggestion, we have made the following changes. (1) We calculated the dominant species of phytoplankton based on Equation 1 (L155) in the manuscript, and judged it as the dominant specie when Y ≥ 0.02, which we have added in Materials and methods (L162). (2) We checked and revised the names of species according to your suggestion, such as Microcystis aeruginosa, Phormidium tenue, Aphanizomenon flos-aquae

line 149: “The number of dominant species after the restoration is more than the restoration,” - I don’t understand that sentence. What did you mean?

Response: We feel sorry for the inconvenience brought to the reviewer. The above sentence means:Before the aquatic vegetation was restored, there were 12 dominant phytoplankton species, after that there were 16 species, and the number of dominant species increased. We have rewritten the above sentence as follows: “Before the restoration of aquatic vegetation, there were 12 dominant phytoplankton species in Caizi Lakes, which grew to 16 species after restoration. In addition, the dominant groups of phytoplankton also changed greatly.”

line 155: “…re mainly cyanobacteria green algae and diatoms…” - mainly cyanobacteria, green algae and diatoms…

Response: Thank for your comments. We are sorry for our oversight. To harmonize the classification of phytoplankton with the above, we replaced cyanobacteria, green algae and diatoms with Cyanophyta, Chlorophyta and Bacillariophyta, respectively.

Discussion

As the results of the study clearly indicate the existence of alternative states in the lake, I believe that the discussion should be conducted in this spirit. I therefore suggest reorganising the discussion and adding one sub-chapter concerning this phenomenon instead just describing the situation found in the lakes.

Response: We gratefully appreciate for your valuable suggestion/ comment. We have added a discussion by reviewing the literature as follows.

4.4 Mechanism of formation of alternative states of the lake

Studies have shown that two different alternative equilibria states exist in shallow lakes, namely a clear state dominated by submerged vegetation and a turbid state dominated by phytoplankton [48]. Ecosystems respond to external changes and perturbations in different ways and may switch from one state to another when external condition thresholds are exceeded, but it can change back to the original state at different thresholds when conditions are reversed [49].

Caizi Lakes are a classic example of a lake that had experienced different alternative states over many years. Caizi Lakes were in pristine state with clear water and abundant submerged vegetation. However, however, with the continued occurrence of productive fish farming disturbances, progressively increasing nutrient loads led to increased turbidity, reduced transparency, and submerged vegetation due to lack of light and thus its growth was limited. This caused a catastrophic change in the lake to a turbid state dominated by phytoplankton due to increased turbidity - through various mechanisms - and loss of vegetation, which was difficult to recover naturally [50].

After the large-scale removal of farming seines, Caizi Lakes changed from a turbid state dominated by phytoplankton to a clear state dominated by aquatic vegetation, and these changes may be attributed to the positive feedback between water transparency and aquatic vegetation. The positive effect of aquatic vegetation, which can improve environmental conditions and improve water clarity for better growth, may help create positive feedbacks. However, this positive feedback effect was more pronounced for submerged vegetation, which has more stringent transparency requirements for growth. Our study showed that after the removal of the seine net, the aquatic vegetation in Caizi Lakes was dominated by floating vegetation rather than submerged vegetation, which is an alternative steady state occurrence. This was significant in that it indicated that there was a hysteresis in the lake during the alternative state transition. It has been shown that a return to a state dominated by pristine, clear and submerged vegetation requires a reduction in nutrient loading to levels well below those that existed when the vegetation collapsed [51]. In addition, increased aquatic vegetation can provide refuge for zooplankton. This increased the risk of phytoplankton predation by zooplankton and also reduced the competitive effects of phytoplankton and aquatic vegetation, thus increasing the growth advantage of aquatic vegetation. Therefore, the change from turbid water to clear water after the removal of the seine may be formed by a positive feedback mechanism between biotic and abiotic factors or biotic and biotic. (L1259-1296)

Thank you again!

Round 2

Reviewer 1 Report

Thank you for your carreful corrections and for having taken into account my numerous comments. I think that the quality of the manuscript has greatly enhanced. My only comment concerns the english language, which could still be improved in my opinion.

Author Response

Respose to the Review 1 Comments.

Dear reviewers,

Re: Manuscript ID: diversity-1671136,

Title: Effect of aquatic vegetation restoration after removal of culture purse seine on phytoplankton community structure in Caizi Lakes

We would like to thank you again for your careful reading, helpful comments and constructive suggestions, which have greatly improved our manuscript.

My manuscript has been perfected by Marci Baun from the University of California, Los Angeles.

Thank you again!

Reviewer 2 Report

I am pleased to say that many of my comments have been taken into consideration. I think the manuscript is in much better shape now. My concern is still that the methods of phytoplankton study are still poorly presented. I mean, for example, a poor presentation of how the abundance was counted, by what method. Also, some phytoplankton species names are outdated and rarely used anymore. Please check them out.

I have some additional comments:

Line 177: „..by nessler’s reagent..”  Should be ..by Nessler’s reagent..

Line 178: by Ultraviolet spectrophotometry - UV spectrometry or ultrafiolet spectrometry.

Line 178 and throughout the text: ”Chl. a” - a correct form is “chl-a”.

Line 187: A name of the counting method of phytoplankton should be introduced in the section of M&M. There is several methods to count phytoplankton cells, so that one you used should be named.

Line 288 and throughout the text: The TLI is sometimes written in italics, sometimes it isn’t so.

Lines 336, 343: Cai zi and Lake Xizi

Line 371 and throughout the text: “Cyanophyta” The correct name for this group of organisms in taxonomy is Cyanobacteria. I insist on using proper names of phytoplankton. There is Cyanophyceae as a class, but phylum is Cyanobateria, not Cyanophyta.

…and Aphanocapsa elachista;… There should be a full stop at the end of a sentence, not a semicolon.

Line 526 and throughout the text: “Melosira granulata” I made a comment concerning current names of species in the first round of the review, but the authors didn’t check them in the available bases. For example, Melosira granulata is now (for many years) Aulacoseira granulata. And phycologists and hydrobiologists don’t use the old name. The same with Anabaena circinalis and some others. Please check the names of species carefully.

Line 589: “S. Quadricauda” - S. quadricauda

Line 764: flosaquae should be flos-aquae

Lines 836-841: “Caizi Lakes are a classic example of a lake that had experienced different alternative states over many years. Caizi Lakes were in pristine state with clear water and abundant submerged vegetation. However, however, with the continued occurrence of productive fish farming disturbances, progressively increasing nutrient loads led to increased turbidity, reduced transparency, and submerged vegetation due to lack of light and thus its growth was limited.” Please check English in this sentence.

Lines 860-862: “Therefore, the change from turbid water to clear water after the removal of the seine may be formed by a positive feedback mechanism between biotic and abiotic factors or biotic and biotic.” Please rewrite this sentence. Especially the end sounds strange.

Author Response

Respose to the Review 2 Comments.

Dear reviewers,

Re: Manuscript ID: diversity-1671136,

Title: Effect of aquatic vegetation restoration after removal of culture purse seine on phytoplankton community structure in Caizi Lakes

We would like to thank you again for your careful reading, helpful comments and constructive suggestions, which have greatly improved our manuscript.

We have carefully considered all comments from the reviewers and revised our manuscript accordingly. We have uploaded a copy of the original manuscript with all the changes highlighted by using the track changes mode in MS Word. Revised portion are marked in red in the paper. The main corrections in the paper and the responds to the reviewer’s comments are as flowing:

Line 177: by nessler’s reagent..” Should be ..by Nessler’s reagent..

Response: We appreciate you pointing out this issue. We have finished it in the revised manuscript.

Line 178: by Ultraviolet spectrophotometry - UV spectrometry or ultrafiolet spectrometry.

Response: We appreciate you pointing out this issue. We have finished it in the revised manuscript.

Line 178 and throughout the text: ”Chl. a” - a correct form is “chl-a”.

Response: We appreciate you pointing out this issue. We think your statement is feasible, but we are sorry that we did not modify it as you requested. Regarding the writing of “Chl. a”, we have reviewed the literature and believe that “Chl. a” is also correct.

Line 187: A name of the counting method of phytoplankton should be introduced in the section of M&M. There is several methods to count phytoplankton cells, so that one you used should be named.

Response: We appreciate you pointing out this issue. We have finished it in the revised manuscript.

Line 288 and throughout the text: The TLI is sometimes written in italics, sometimes it isn’t so.

Response: We appreciate you pointing out this issue. We have checked the italics of the words.

Lines 336, 343: Cai zi and Lake Xizi

Response: We appreciate you pointing out this issue. We have finished it in the revised manuscript.

Line 371 and throughout the text: “Cyanophyta” The correct name for this group of organisms in taxonomy is Cyanobacteria. I insist on using proper names of phytoplankton. There is Cyanophyceae as a class, but phylum is Cyanobateria, not Cyanophyta. …and Aphanocapsa elachista;… There should be a full stop at the end of a sentence, not a semicolon.

Response: We appreciate you pointing out this issue. We modified Cyanophyta to Cyanobacteria in the full text and modified the punctuation marks

Line 526 and throughout the text: “Melosira granulata” I made a comment concerning current names of species in the first round of the review, but the authors didn’t check them in the available bases. For example, Melosira granulata is now (for many years) Aulacoseira granulata. And phycologists and hydrobiologists don’t use the old name. The same with Anabaena circinalis and some others. Please check the names of species carefully.

Response: We appreciate you pointing out this issue. We have finished it in the revised manuscript. Anabaena circinalis modify to Dolicospermum circinale, Ankistrodesmus angustus modify to Monorapidium contortum, Pediastrum simplex modify to Monactinus simplex, Synedra acus modify to Ulnaria acus, Cymbella perpusilla modify to Encyonema perpusillum.

Line 589: “S. Quadricauda” - S. quadricauda

Response: We appreciate you pointing out this issue. We have finished it in the revised manuscript.

Line 764: flosaquae should be flos-aquae

Response: We appreciate you pointing out this issue. We have finished it in the revised manuscript.

Lines 836-841: “Caizi Lakes are a classic example of a lake that had experienced different alternative states over many years. Caizi Lakes were in pristine state with clear water and abundant submerged vegetation. However, however, with the continued occurrence of productive fish farming disturbances, progressively increasing nutrient loads led to increased turbidity, reduced transparency, and submerged vegetation due to lack of light and thus its growth was limited.” Please check English in this sentence.

Response: We appreciate you pointing out this issue. We have finished it in the revised manuscript. Our modifications are as follows: “Caizi Lakes are typical shallow lake that undergone different alternative states over the years. Under original conditions, Caizi Lakes had clear water quality and abundant submerged vegetation. However, with the rapid development of aquaculture in Caizi Lakes, the water quality had greatly reduced, such as increased turbidity and reduced transparency of water column.”

Lines 860-862: “Therefore, the change from turbid water to clear water after the removal of the seine may be formed by a positive feedback mechanism between biotic and abiotic factors or biotic and biotic.” Please rewrite this sentence. Especially the end sounds strange.

Response: We appreciate you pointing out this issue. We have finished it in the revised manuscript. Our modifications are as follows: In summary, after the removal of the seine, Caizi Lakes changed from a turbid state dominated by phytoplankton to a clear state dominated by aquatic vegetation, and this change may be related to the positive feedback between aquatic vegetation and water transparency, and may also be caused by the interaction between aquatic vegetation and aquatic organisms.

Thank you again!
